# Rhythmic coordination and ensemble dynamics in the hippocampal-prefrontal network during odor-place associative memory and decision making

**Claire A Symanski[1†], John H Bladon[1,2†], Emi T Kullberg[1,2], Paul Miller[1,3], Shantanu P Jadhav[1]\***

[1]Neuroscience Program, Brandeis University, Waltham, United States; [2]Department of Psychology, Brandeis University, Waltham, United States; [3]Volen National Center for Complex Systems, Brandeis University, Waltham, United States

**Abstract** Memory-guided decision making involves long-range coordination across sensory and cognitive brain networks, with key roles for the hippocampus and prefrontal cortex (PFC). In order to investigate the mechanisms of such coordination, we monitored activity in hippocampus (CA1), PFC, and olfactory bulb (OB) in rats performing an odor-place associative memory guided decision task on a T-maze. During odor sampling, the beta (20–30 Hz) and respiratory (7–8 Hz) rhythms (RR) were prominent across the three regions, with beta and RR coherence between all pairs of regions enhanced during the odor-cued decision making period. Beta phase modulation of phase-locked CA1 and PFC neurons during this period was linked to accurate decisions, with a key role of CA1 interneurons in temporal coordination. Single neurons and ensembles in both CA1 and PFC encoded and predicted animals' upcoming choices, with different cell ensembles engaged during decision-making and decision execution on the maze. Our findings indicate that rhythmic coordination within the hippocampal-prefrontal-olfactory bulb network supports utilization of odor cues for memory-guided decision making.

## Editor's evaluation

The authors report coordination mechanisms between oscillations recorded in the CA1 subfield of the hippocampus, prefrontal cortex, and olfactory bulb and cell ensemble activity in CA1 and prefrontal cortex during odor-cued decision-making. The important findings support the hypothesis that the β rhythm plays a role in coordinating CA1-prefrontal cortex ensembles during decision-making. Sensory-guided decision-making is of broad significance to many readers who are studying executive functions and decision-making behaviors, and the observations reported in this manuscript provide convincing evidence of mechanisms that may support these functions and behaviors.

## Introduction

The ability to recall associations from memory and use them to guide behavior is a key aspect of cognition across species. Animals can associate sensory cues in the environment with rewarding and noxious experiences and utilize these cues for adaptive behavior. Memory-guided decision making demonstrates the brain's remarkable ability to link familiar cues with actions and beneficial outcomes, but little is known about the mechanisms responsible for this cognitive function.

**\*For correspondence:**
shantanu@brandeis.edu

[†]These authors contributed equally to this work

**Competing interest:** The authors declare that no competing interests exist.

Neurons that encode learned associations and reflect upcoming choice behavior have been reported in multiple regions in different sensory modalities (*Allen et al., 2016*; *Fitzgerald et al., 2011*; *Harvey et al., 2012*; *Igarashi et al., 2014*; *Johnson and Redish, 2007*; *McKenzie et al., 2014*; *Moita et al., 2003*; *Otto and Eichenbaum, 1992b*; *Schoenbaum and Eichenbaum, 1995b*; *Shadlen and Newsome, 2001*; *Singer et al., 2013*; *Wirth et al., 2009*; *Wirth et al., 2003*; *Yanike et al., 2004*). However, sensory cued decision-making based on learned associations necessarily involves a brain-wide network that links primary sensory areas, the medial temporal lobe, and higher cortical areas involved in executive function. Numerous studies have highlighted the significance of the hippocampus and prefrontal cortex (PFC) in cognitive processing related to memory and decision making (*Battaglia et al., 2011*; *Euston et al., 2012*; *Floresco et al., 1997*; *Lee and Solivan, 2008*; *Miller and Cohen, 2001*). Both regions are known to encode behaviorally relevant cues and task features (*Gothard et al., 1996*; *Hyman et al., 2012*; *Wiener et al., 1989*; *Wirth et al., 2009*; *Wirth et al., 2003*; *Yanike et al., 2004*), and have been shown to play key roles in memory recall (*Fortin et al., 2004*; *Hasegawa, 2000*; *Siegle and Wilson, 2014*; *Wiltgen et al., 2004*). Notably, coordinated activity between the hippocampus and PFC, supported by bidirectional anatomical connections (*Cenquizca and Swanson, 2007*; *Delatour and Witter, 2002*; *Ito et al., 2015*), has been shown to be critical for learning and memory-guided behavior (*Maharjan et al., 2018*; *Place et al., 2016*; *Shin and Jadhav, 2016*; *Shin et al., 2019*; *Yu and Frank, 2015*; *Zielinski et al., 2020*). Therefore, we focused on the coordinated interactions between hippocampus and PFC as a potential key mechanism through which learned associations are recalled and translated into memory-guided decisions.

Several studies have shown that rhythmic network oscillations in the local field potential (LFP) are involved in long-range interactions between the hippocampus and PFC (*Benchenane et al., 2011*; *Buzsáki and Draguhn, 2004*; *Colgin, 2011*; *Gordon, 2011*; *Jones and Wilson, 2005*; *Shin and Jadhav, 2016*). In particular, phase coherence in distinct frequency bands across this network has been suggested as a mechanism for network coordination underlying mnemonic functions. Notably, hippocampal-prefrontal coherence in the theta rhythm (6–12 Hz) and phase-locked spiking plays a role in spatial working memory and acquisition of spatial tasks (*Benchenane et al., 2010*; *Gordon, 2011*; *Hyman et al., 2010*; *Jones and Wilson, 2005*). However, whether similar mechanisms of coordination between hippocampus and PFC underlie decision making based on sensory cued associations is unclear.

Rodents rely heavily on odor cues for navigation and foraging, and odor memories are highly salient and robust (*Abraham et al., 2004*; *Eichenbaum, 1998*; *Rinberg et al., 2006*; *Uchida and Mainen, 2003*), making olfactory memory tasks ideally suited for studying memory-guided decision making. Previous studies using odor memory tasks have found prominent beta (20–30 Hz) oscillations in olfactory regions and the medial temporal lobe during cue sampling, suggesting that the beta rhythm acts as a potential mode of long-range communication for olfactory information processing (*Frederick et al., 2016*; *Igarashi et al., 2014*; *Kay and Beshel, 2010*; *Rangel et al., 2016*; *Stopfer et al., 2003*). In addition to the beta rhythm, the respiratory rhythm (RR; 7–8 Hz), driven by the animal's breathing cycle, is also prominent in the hippocampus during mnemonic processing of odor stimuli (*Karalis and Sirota, 2022*; *Kay, 2005*; *Kepecs et al., 2006*; *Lockmann et al., 2016*; *Nguyen Chi et al., 2016*; *Verhagen et al., 2007*). However, not much is known about the roles of these rhythms in coordinating activity in the hippocampal-prefrontal network during odor-cued decision making.

To elucidate these mechanisms, we employed an odor-place association task in which rats were required to choose the correct trajectory on a T-maze by recalling and utilizing familiar associations between odor cues and reward locations. While rats were performing the task, we recorded simultaneously from the hippocampus and PFC. Given the involvement of both of these regions in odor-memory tasks (*Alvarez et al., 2002*; *Eichenbaum et al., 1986*; *Fujisawa et al., 2008*; *Fujisawa and Buzsáki, 2011*; *Martin et al., 2007*; *Otto and Eichenbaum, 1992a*; *Peters et al., 2013*; *Place et al., 2016*), and the role of hippocampal-prefrontal networks in memory-guided behavior (*Churchwell et al., 2010*; *Fortin et al., 2004*; *Hasegawa, 2000*; *Moser and Moser, 1998*; *Wiltgen et al., 2004*; *Wiltgen et al., 2010*), we hypothesized that rhythmic activity in this network may govern the cellular representation of odor-cued decisions underlying behavioral choices. We also monitored LFP activity in the olfactory bulb (OB) in addition to CA1 and PFC, for robust samplings of previously reported olfactory rhythms such as beta and RR, which have been implicated in cognitive processing of olfactory stimuli (*Kay, 2014*; *Kay et al., 2009*). Our results point to a role of beta and RR rhythms in

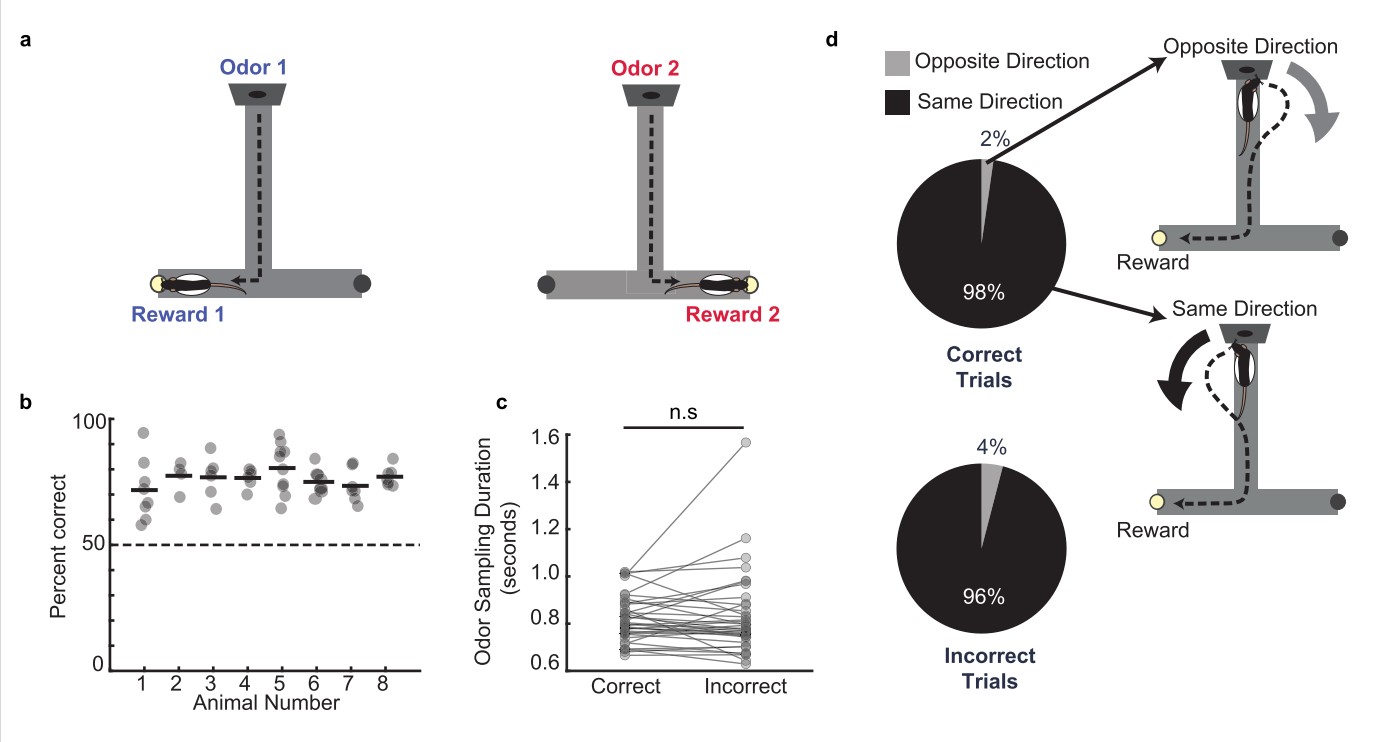

**Figure 1.** Odor–place associative memory and decision-making task. (**a**) Schematic of the odor-cued T-maze task. Odors 1 (heptanol) and 2 (ethyl butyrate) were delivered at the odor port in pseudo-random order. Presentation of Odors 1 and 2 was associated with milk reward at Reward locations 1 and 2, respectively. Animals had to recall odor-space associations on each trial and utilize the association to choose the correct reward location. (**b**) Performance of each animal (n = 8) on the odor-cued T-maze task (animals 1–5) or truncated odor-cued task (animals 6 - 8; truncated task, without spatial delay period) across multiple sessions (grey dots). Animal average is indicated by black bars. Dashed line indicates chance level. (**c**) Odor sampling duration across all sessions (n = 38) on correct and incorrect trials (signed-rank test, p = 0.71). (**d**) Turn direction away from odor port in relation to chosen reward well. Pie charts indicate the fraction of trials in which, at the odor port, animals turned in the same direction versus the opposite direction as the reward well that they would ultimately choose. Sessions in which animals ran the truncated task without spatial delay period were excluded (correct trials: n = 1,624 same direction, 32 opposite direction, binomial test, p = 0; incorrect trials: n = 499 same direction, 21 opposite direction, p = 1.4e-108).

The online version of this article includes the following figure supplement(s) for figure 1:

**Figure supplement 1.** Behavioral parameters on the odor-place associative memory and decision-making task.

coordinating the olfactory-hippocampal-prefrontal network for utilizing learned odor-place associations to inform decisions, and shed light on the cellular and network mechanisms underlying this process.

## Results

### Odor-place associative memory and decision-making task

The odor-cued spatial associative memory task required rats to sniff at an odor port where one of two possible odors was presented using a calibrated olfactometer (see **Methods**). The rats were required to choose the correct associated reward arm on the T-maze based on the sampled odor identity, where they would receive a reward of evaporated milk (**Figure 1a**).

Rats were habituated to the maze and pre-trained on the task. Rats were initially shaped using blocked trials on a truncated maze before progressing to the long-stem T-maze (see **Methods**). After reaching criterion (80% performance for 3 consecutive days), animals were surgically implanted with a tetrode microdrive array for recording neural data. Recordings in three animals were performed only on the truncated maze without the long center stem (spatial delay) (**Methods,** animals 6–8). Following post-operative recovery and during electrophysiological recording, animals maintained a high level of performance on the task, indicating accurate decision making based on cued recall of odor-place

associations (*Figure 1b*, n=8 rats, 77.0% ± 1.3%, mean ± s.e.m.). Rats were required to hold their nose in the odor port for a minimum of 0.50 s on each trial, but could continue sniffing the odor for any length of time after the minimum threshold was reached. The odor was continuously dispensed for the entire duration of time that the rat held its nose in the odor port and was only turned off once the rat disengaged from the odor-port. The average odor-sampling duration before odor port disengagement and odor offset was 0.82±0.02 s (mean ± s.e.m. across sessions), and this duration was similar between correct and incorrect trials (*Figure 1c*; within-session signed-rank test, p=0.71, distribution for all trials shown in *Figure 1—figure supplement 1a*). The animals' average velocity during the task showed a decrease in speed from the pre-odor period to the odor sampling period in the odor port, followed by an increase in speed after they left the odor port to run to the reward location (*Figure 1—figure supplement 1b–c*). We observed rapid movement away from the odor-port after odor-port disengagement (*Figure 1—figure supplement 1d*). In two animals, a thermocouple was implanted in the nasal cavity to measure the sniff rhythm (see **Methods**). There was a small but significant increase in sniff rate during the odor sampling period (within-session increase 7.1±0.39 Hz, mean ± s.e.m.) compared to time matched pre-odor periods (6.2±0.29 Hz, mean ± s.e.m.) (*Figure 1—figure supplement 1e*).

Notably, we observed that on a majority of trials (95.4% ± 0.12%), the animals' turn direction away from the odor port matched the direction of the T-maze reward arm that they would ultimately choose on that trial. This behavioral phenomenon was not required for successful performance of the task, and it occurred regardless of whether the trial was correct or incorrect (*Figure 1d*; within-session binomial tests, correct trials: p=0; incorrect trials: p=1.4e-108). This observation indicates that the rats recall the odor-place association and choose the reward location for each trial *during* the odor sampling period, before exiting the odor port to run toward the reward. The time of disengagement from the odor port thus provides a trial-by-trial estimate of the moment at which the animal executes the decision. The odor sampling period thus corresponds to odor-cued recall of the learned association and priming of the subsequent decision to turn toward the reward location, with a behavioral report of the decision occurring at odor port offset. We therefore termed this odor sampling period as 'the decision-making period', since it provides a temporal window between odor onset and odor port exit to investigate mechanisms underlying odor-cued decision making.

## Beta and RR coherence is elevated during odor sampling and decision making

We first focused on the decision-making period and first sought to determine the network dynamics that underlie coordination of brain regions during this period. We used a tetrode microdrive array to record local field potentials (LFPs) and single units from the dorsal CA1 region of the hippocampus, the prelimbic region of the prefrontal cortex (PFC, primarily prelimbic area), and the olfactory bulb (OB, only LFP) in rats as they performed the odor-place association task (see **Methods;** *Figure 2—figure supplement 1a–b*). The thermocouple signal and LFP traces from CA1, PFC, and OB from an example trial are shown in *Figure 2a*, along with the same LFP signals filtered in the 20–30 Hz band (additional example shown in *Figure 2—figure supplement 1c*).

We observed a strong increase in power in the beta band (20–30 Hz) during this decision-making period compared to a time-matched pre-odor period across all three regions (*Figure 2b and d*, Wilcoxon signed-rank test, n=38 sessions, CA1 *P*=2.84e-3, PFC *P*=1.80e-4, OB *P*=8.3e-6). Power spectra aligned to odor offset are shown in *Figure 2d* (alignment to odor onset in *Figure 2—figure supplement 2d*). Similar increases in beta power during odor discrimination tasks have been reported previously in OB, CA1, and lateral entorhinal cortex (*Frederick et al., 2016*; *Igarashi et al., 2014*; *Kay and Beshel, 2010*; *Rangel et al., 2016*). The respiratory rhythm (RR, 7–8 Hz) was also prominent in the LFP in all three regions but did not increase significantly following odor onset (*Figure 2d*). This rhythm, which corresponds to the respiration rate during odor sampling, has previously been shown to be physiologically and mechanistically distinct from the 6–12 Hz hippocampal theta rhythm (*Lockmann et al., 2016*; *Nguyen Chi et al., 2016*), although there is overlap between the two frequency bands. Following odor port disengagement and the initiation of running down the track, we observed a small shift in the dominant LFP frequency from RR to the theta band in CA1 and PFC, reflecting the change in behavioral state (*Figure 2—figure supplement 1e*).

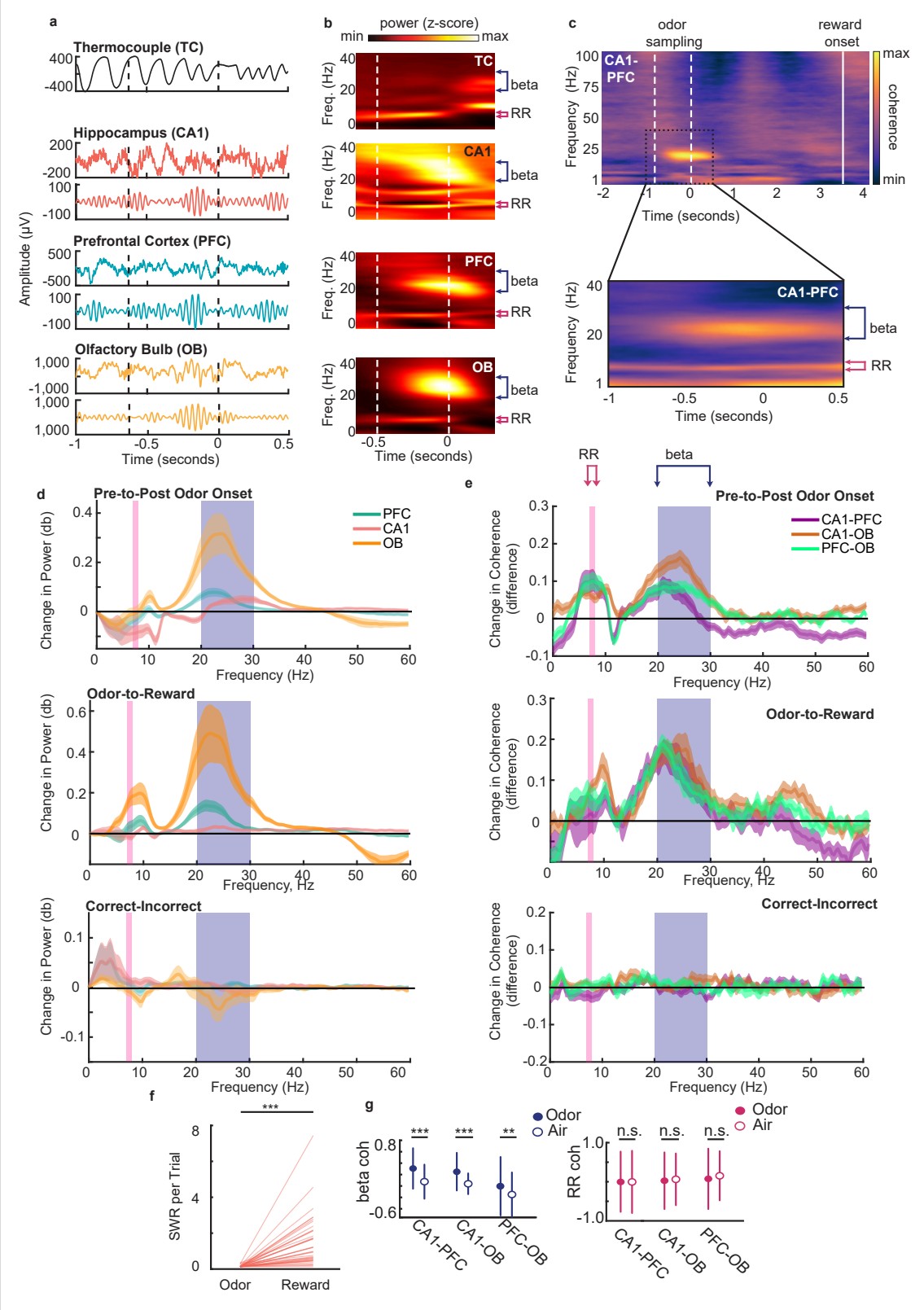

**Figure 2.** Beta rhythms support decision making based on odor-place associations. (**a**) Examples of thermocouple and LFP traces from one tetrode in each region during presentation of odor from one trial, aligned to odor port disengagement. Area between dashed lines indicates odor sampling period. Top to bottom: Respiratory rhythm recorded via thermocouple, CA1 signal, beta band (20–30 Hz) filtered CA1 signal, PFC signal, beta band filtered PFC signal, OB signal, beta band filtered OB signal. (**b**) Time-frequency plot showing power spectra aligned to odor offset. Color scale

*Figure 2 continued on next page*

*Figure 2 continued*

represents z-scored power. Area between dashed lines indicates average odor sampling period. Beta band is marked by blue bracket (20–30 Hz). RR band is marked by pink bracket (7–8 Hz). Thermocouple signal (TC), n = 12 sessions, max 0.69, min –0.31; CA1: n = 39 sessions, max 0.26, min –0.35; PFC: n = 38 sessions, max 0.60, min –0.16; and OB: n = 39 sessions, max 2.17, min –0.24. (**c**) Top: CA1-PFC coherence spectra for one animal during the full task time window from odor sampling (area between white dashed lines, aligned to odor offset) to average reward onset time (solid line) (n = 4 sessions, max 0.84; min –0.41). Bottom: CA1-PFC coherence aligned to odor offset across all animals (n = 38 sessions, max 0.51; min –0.34). Color scale: z-scored coherence. (**d**) Top: Change in PFC, CA1 and OB LFP power from pre-to-post odor onset (signed-rank tests, n = 38 sessions, Beta band: CA1 p=2.84e-3, PFC p=1.80e-4, OB p=8.3e-6, RR band: CA1 [decrease] p=3.94e-3, PFC p=0.551, OB p=0.949). Middle: Change in LFP Power from odor-to-reward period (signed-rank tests, Beta: sign rank tests, CA1 p=0.0012, PFC p=5.3e-4, OB p=4.4e-4, RR: CA1 p=0.11, PFC p=0.062, OB p=4.4e-4). Bottom: change in LFP power from incorrect-to-correct trials (trial-count matched, ranksum test on session, no difference were found for either band). Shaded regions represent SEM. (**e**) Top: change in coherence between region-pairs from pre-to-post odor onset (signed-rank test, n=38 sessions, Beta: CA1-PFC p=2.84e-3, CA1-OB p=5.1e-5, PFC-OB p=3.67e-5; RR: CA1-PFC p=3.94e-3, CA1-OB p=6.35e-4, PFC-OB p=2.43e-4). Middle: change in coherence between region-pairs from reward period to odor sampling period (signedrank tests, Beta; CA1-PFC p-0.01, CA1-OB 0=0.0012, PFC-OB p=0.0024, RR: CA1-PFC p=0.57, CA1-OB p=0.0044, PFC-OB p=0.64e-4).Bottom: change in coherence from incorrect-to-correct trials (trial-count matched, ranksum test on sessions, no significant difference were found for either band). Shaded regions represent SEM. (**f**) Number of SWR events per trial during odor sampling and reward consumption on correct trials (signed-rank test, n = 38 sessions, p = 1.1e-7***). (**g**) Left: Beta Z-scored coherence between CA1-PFC, CA1-OB and PFC-OB on correct trials with odors (n = 3174) and randomly rewarded trials with only air presented at the odor port (n = 134). Right: same as left, but for RR. Correct trials were randomly subsampled to match the number of incorrect trials 1000 times. Error bars indicate s.d. (Bootstrap tests: beta: CA1-PFC: p < 0.001***, CA1-OB, p < 0.001***, PFC-OB: p = 0.002**; RR: CA1-PFC: p = 0.49, CA1-OB, p = 0.30, PFC-OB, p = 0.12).

The online version of this article includes the following figure supplement(s) for figure 2:

**Figure supplement 1.** Rhythmic activity in the olfactory-hippocampal- prefrontal network.

**Figure supplement 2.** Beta coordination underlies decision making based on odor-place associations.

Odor sampling during the decision-making period thus drives prominent increases in beta rhythm power in OB, CA1, and PFC. During the decision-making period, we also found phase-amplitude coupling between beta and RR rhythms in all three regions, consistent with previous findings (*Lockmann et al., 2016*; *Figure 2—figure supplement 1f*)**,** and suggesting a mechanistic relationship between the two rhythms. Despite this relationship, previous literature shows that the two rhythms are differentially generated and maintained in the olfactory system and hippocampus and appear to serve different functions in olfactory processing (*Kay, 2014*; *Kay et al., 2009*; *Martin et al., 2006*; *Neville and Haberly, 2003*), indicating that the beta rhythm is not merely a harmonic of RR. Indeed, power increases were seen during the odor sampling period only in the beta band (*Figure 2d* signed-rank tests, n=38 sessions, CA1 p=2.84e-3, PFC p=1.80e-4, OB p=8.3e-6) and not in the RR band (*Figure 2d* signed-rank tests, n=38 sessions, CA1 [decrease in power] p=3.94e-3, PFC p=0.551, OB p=0.949).

The hippocampal-prefrontal-olfactory bulb (CA1-PFC-OB) network also exhibited prominent and consistent coherence at the beta frequency band during odor sampling, leading into decision execution at odor-port offset (*Figure 2c* for CA1-PFC**,** similar figures for CA1-OB and PFC-OB in *Figure 2—figure supplement 2b–c*). This coherence was specific to the odor-sampling period: it significantly increased compared to time-matched pre-odor periods (*Figure 2e top*, signed-rank test, n=38 sessions, CA1-PFC p=2.8e-3, CA1-OB p=5.1e-5, PFC-OB p=3.7e-5), and diminished shortly after the rat exited the odor port, once the decision had been made (*Figure 2c*). Network wide coherence in the RR band also increased from the pre-odor to the odor-sampling period (*Figure 2e top***,** signed-rank test, CA1-OB p=6.4e-4, CA1-PFC p=3.9e-3, PFC-OB p=2.4e-4). During running, after the decision-making period and exit from the odor-port, the prominent low frequency RR coherence shifted to be slightly higher and centered on the canonical theta band (6–12 Hz), similar to what was observed in the LFP power spectrum (*Figure 2—figure supplement 2a*).

To control for the possibility that elevated beta coherence during the decision-making period may simply be a reflection of passive movement preparation, as has been observed in sensorimotor cortex (*Donoghue et al., 1998*), we compared CA1-PFC-OB beta coherence during the last 500ms of the decision-making period with the last 500ms just prior to reward well exit. We confirmed that beta coherence observed during the decision-making period was significantly higher than coherence at the reward well between all three pairs (*Figure 2e middle*, signed-rank test, CA1-PFC p=0.010, CA1-OB p=0.0012, PFC-OB p=0.0024). Interestingly, this effect in RR was observed for CA1-OB and PFC-OB, but not between CA1-PFC (*Figure 2e middle*, signed-rank test CA1-PFC p=0.569, CA1-OB

p=0.0044, PFC-OB p=0.00064), suggesting that in contrast to beta coherence, the increase in CA1-PFC RR coherence was not specific to the decision making period. In an additional control, we also confirmed that this elevated beta coherence was specifically seen during odor-cued decision making and was not present in sessions in which only air was presented at the odor-port (*Figure 2g*), similar to previous reports for beta coherence in CA1-entorhinal cortex (*Igarashi et al., 2014*) (see **Methods** for description of air-only sessions). In these uncued air-only sessions performed after the rat learned the odor task, reward was given on random trials at the reward locations, even though no odors were presented (*Figure 1—figure supplement 1f*). RR coherence was unchanged between odor- and air-cued sessions, but beta coherence was significantly lower in the absence of odor cues during air-only sessions (*Figure 2g*).

Next, since hippocampal sharp-wave ripples (SWRs) have been proposed as a mechanism for memory retrieval, decision making, and planning in some tasks (*Carr et al., 2011*; *Joo and Frank, 2018*; *Norman et al., 2019*; *Singer et al., 2013*), we investigated the occurrence of SWRs during our task. SWR events occurred frequently at the two reward locations, as expected (*Buzsáki et al., 1983*). However, a distinct lack of SWR events at the odor port (*Figure 2f*; signed-rank test, p=1.1e-7) suggests that SWR replay is unlikely to play a direct role in decision making based on recalled associative memories for this well-learned task.

CA1-PFC-OB beta coherence was thus specifically enhanced during the odor sampling and decision making period, suggesting that beta rhythms can play a role in odor-cued decision making in this task. The increase in RR also suggests a possible role in organizing sensory information transfer in odor tasks, as suggested by previous studies (*Kay, 2005*; *Kepecs et al., 2006*; *Nguyen Chi et al., 2016*). We also examined phase differences in beta and RR oscillations between all pairs of regions during decision-making (*Figure 2—figure supplement 2d*). Although there is a slight preference for CA1 leading PFC for beta and RR rhythms, the phase offsets across regions showed high variability across animals and sessions, presumably due to differences in electrode locations across animals, especially in PFC and OB, precluding an interpretation of directionality of interactions.

We next investigated the relationship between beta and RR power and coherence and performance on the task. There was no significant change in the level of coherence or power in the CA1-PFC-OB network in either frequency band between correct and incorrect trials (*Figure 2e bottom*, signed-rank tests; all p's>0.05). Therefore, the strength of overall oscillatory coordination in the network as measured by coherence and power may not directly enable a correct decision. This result leaves open the possibility that oscillatory phase modulation of neuronal activity may instead play a role.

## Single neurons in CA1 and PFC exhibit choice selectivity during decision-making period

In addition to LFP, we assessed neuronal activity in the task by recording single units from hippocampal area CA1 (n=1,309 units) and PFC (n=717 units) (distribution of neurons across animals shown in table in *Supplementary file 1*). Many of these units fired too sparsely on the maze or only during sleep, and were excluded from further analyses. Therefore we first selected units that fired at least 100 spikes during the run epochs (*Figure 3g–h*, 'Active Cells', 934 in CA1, 508 in PFC) and categorized them into pyramidal cells and interneurons based on firing rate and spike width (see **Methods,** CA1: 813 (87%) pyramidal, 121, (13%) interneurons; PFC: 464 (91%) pyramidal, 44 (9%) interneurons). For analyses during the decision-making period, we then selected for cells that were active during the odor period with the criterion that they fired at least as many spikes as there were trials ('Odor Period Active', see **Methods,** CA1: 170/813,21% PFC: 234/464, 50%). The majority of these active cells were also task responsive and exhibited significant changes in firing rate following the onset of odor sampling (see Methods and *Figure 3g–h*; n=138/170, 81% task-responsive CA1 cells, n=185/234, 79%, task-responsive PFC cells).

A subset of task-responsive cells was selective for specific choices, based on the identity of the odor and the associated choice. Selectivity was calculated by generating a selectivity index (SI), in which the difference between the average firing rate response to each odor on correct trials was divided by the sum of the two responses, giving a value between –1 and 1 (see **Methods**). To determine significance, the SI value for each task responsive pyramidal cell was compared to a distribution of SIs generated by shuffling the odor identities across trials (n=47/138, 34% CA1 selective cells, n=59/185, 31% PFC selective cells) (*Figure 3* and *Figure 3—figure supplement 1*). These fractions are consistent with

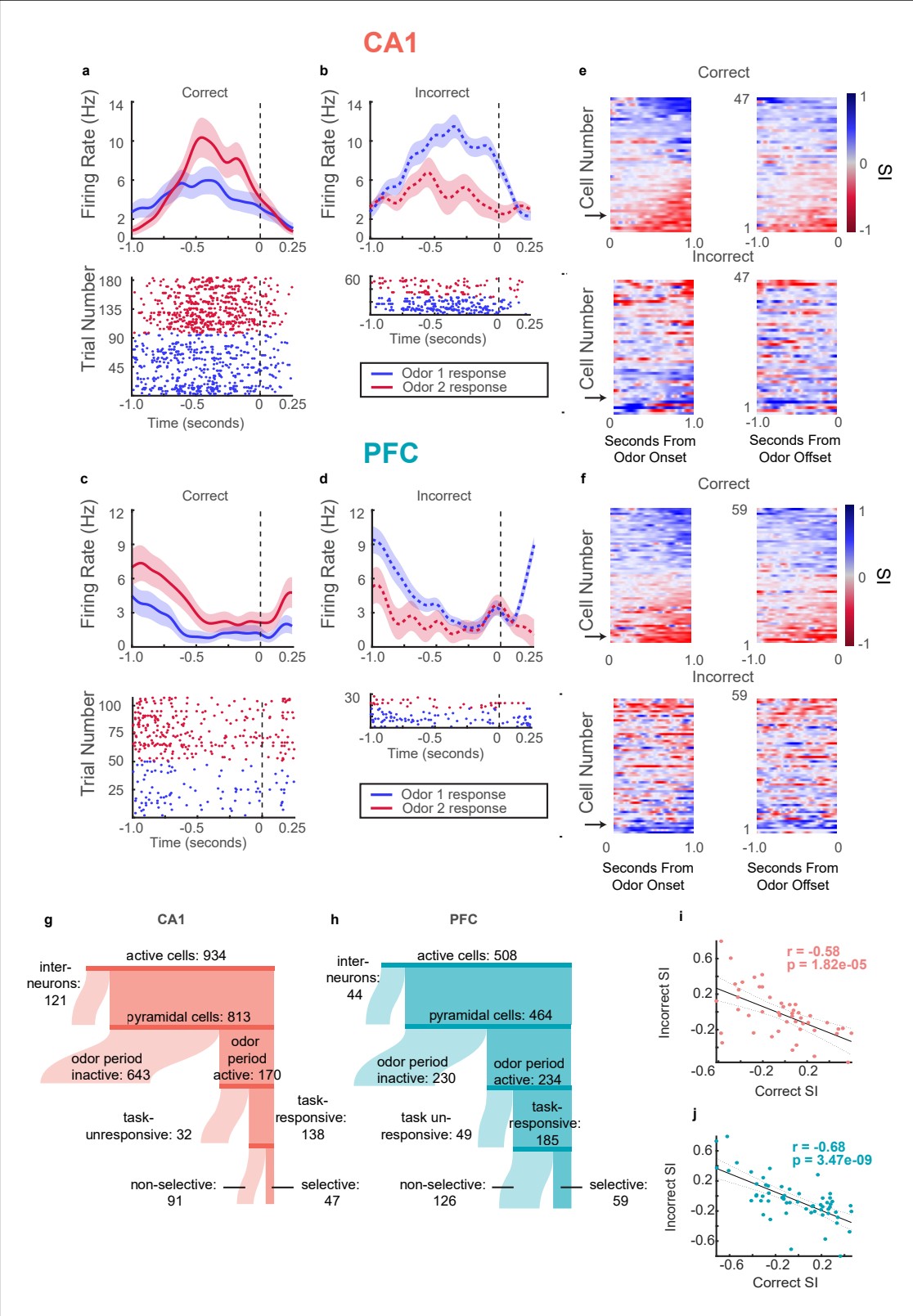

**Figure 3.** Single neurons in CA1 and PFC exhibit choice-selectivity during decision-making period. (**a**) Example PSTH and raster plot for a single choice-selective CA1 neuron on correct trials, aligned to odor-port disengagement. Shaded areas indicate s.e.m. Firing rates are shown in Hz, referring to spikes/second. (**b**) Same as for a, but for incorrect trials. (**c**) Same as a but for PFC cell. (**d**) Same as C but for incorrect trials. (**e**) Selectivity index (SI) of all choice-selective cells in CA1 on correct trials (top) and incorrect trials (bottom), aligned to odor-port engagement(left) and odor-port

*Figure 3 continued on next page*

*Figure 3 continued*

disengagement(right). SI is calculated as the difference in firing rate between Odor 1 trials and Odor 2 trials, divided by the sum of these firing rates. SI is color coded, where blue indicates an SI of 1 (absolute Choice 1 preference), red indicates an SI of -1 (absolute Choice 2 preference) and grey indicates an SI of 0 (not selective). Cells are sorted according to peak selectivity on correct trials and sorting order is the same for both plots. (**f**) Same as E, but for all PFC choice-selective cells. (**g**) Sankey diagram showing the number of CA1 pyramidal cells that were classified into different categories, sizes of partitions are proportional to raw numbers. (**h**) Same as g, but for PFC cells. (**i**) Correlation between Correct trial SI and incorrect trial SI for CA1 selective cells (n=47, p=0.1.82e-5). (**j**) Same as for I, but for PFC cells (n=59, p=3.47e-9).

The online version of this article includes the following figure supplement(s) for figure 3:

**Figure supplement 1.** Neural responses during decision-making period.

other studies examining single unit responses to odor stimuli in these regions (*Otto and Eichenbaum, 1992b*; *Schoenbaum and Eichenbaum, 1995a*; *Taxidis et al., 2020*).

Peri-stimulus time histograms and raster plots aligned to decision time for one example selective cell for each CA1 and PFC are shown in *Figure 3a and c* respectively. Notably, we found that on incorrect trials, selective cells often exhibited responses to the two odors that were opposite to the responses on correct trials (examples shown in *Figure 3b and d*). We found that this phenomenon held true across the population of selective pyramidal cells, such that overall, the SIs on correct trials were anti-correlated with the SIs on incorrect trials (*Figure 3e–f* for tuning curves locked to odor onset (left) and offset (right) on correct (top) and incorrect (bottom) trials, and also *Figure 3i and j* for selectivity scores during correct versus incorrect trials, Spearman Rank Correlation, CA1 pyr correct-incorrect SI $r$=–0.58, p(47) = 1.82e-5, PFC pyr correct-incorrect SI $r$=–0.68, p(59) = 3.47e-9). Note that since there are only two possible choices on the task, the behavioral response on an incorrect trial is identical to that of a correct trial of the opposite odor. The fact that the neural responses on trials with identical behavioral responses are similar indicates that selective cells are not simply coding the odor identity but are instead responsive to the animal's memory-guided decision and upcoming behavioral choice. In contrast, selectivity to the odor identity alone would result in similar responses to a given odor regardless of the upcoming behavior or the ultimate trial outcome. We therefore termed these cells choice-selective cells. This pattern was not significant within the population of neurons that were task-responsive but did not meet the selectivity threshold (*Figure 3—figure supplement 1d–e*). Histograms showing distribution of absolute selectivity indices between selective and nonselective cells are shown in *Figure 3—figure supplement 1f*.

Putative interneurons were also divided into task-responsive and task-unresponsive, using the same criteria as for putative pyramidal cells (*Figure 4a*). *Figure 4b–c* shows interneuron odor selectivity indices and response properties during correct and incorrect trials. For CA1 interneurons, the selectivity response showed anti-correlation during correct versus incorrect trials similar to that of pyramidal cells (*Figure 4c* **top**, r(32) = –0.80, p=3.4e-8) suggesting coordination between these cell populations as well as a functional role in the decision making period. PFC interneurons' SIs on correct and incorrect trials were not anti-correlated, although this could result from the overall low number of selective PFC interneurons (*Figure 4c* **bottom** r(10) = –0.38, p=0.278).

To investigate whether there was evidence of coordination between ensembles in CA1 and PFC, we wanted to determine whether the spiking of neuronal ensembles in the two regions were temporally linked (*Jadhav et al., 2016*; *Siapas et al., 2005*). In order to examine this, we computed the normalized spiking cross-correlation for all CA1-PFC pairs of task-responsive neurons during odor sampling (*Figure 4d*, see **Methods**). Significant peak time lags of cross-correlations were quantified for all CA1-PFC task-responsive putative pyramidal cells (n=224 pairs), all pairs of task-responsive interneurons (n=87 pairs), as well as pairs of CA1 interneurons and PFC pyramidal cells (n=283 pairs), and vice versa (n=111 pairs). We found a significant skew in CA1-PFC interneuron pairs towards the PFC interneurons leading (sign rank test, x=–0.028, p=3.5e-8) (*Figure 4d*), as well as for CA1 pyramidal cell - PFC interneuron pairs in the same direction (sign rank test, x=–0.013, p=0.0034). Additionally, we found a large cluster of CA1 interneuron- PFC pyramidal cell pairs whose cross-correlations peaked around –0.035 s and one at +0.01 s, or approximately the period of a beta cycle. These results further illustrate temporal relationships during the decision making period, and suggest that CA1 interneurons show temporal coordination with PFC task-responsive ensembles in the beta range.

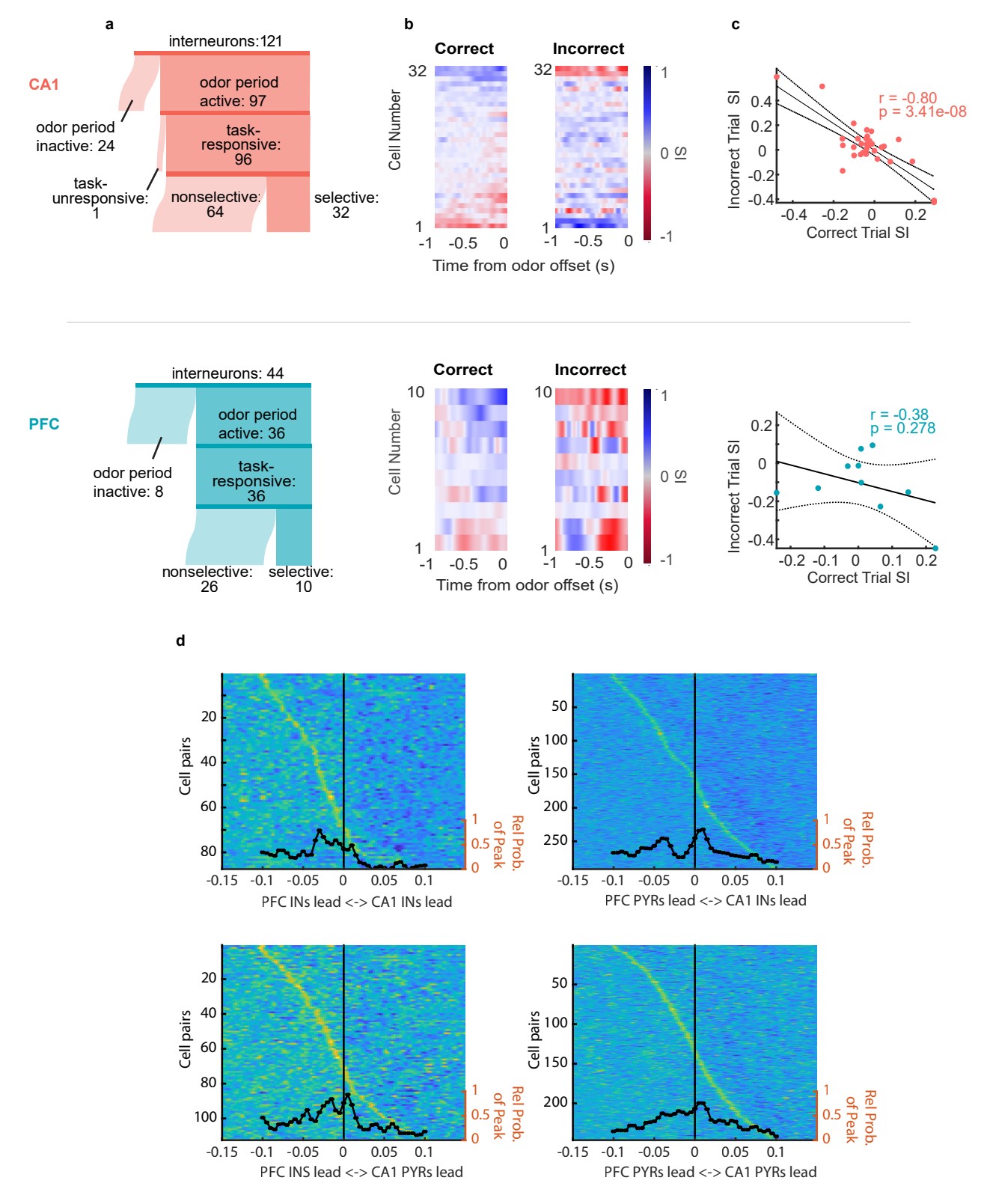

**Figure 4.** Interneurons exhibit choice-selectivity and temporal coordination during decision- making period. (**a**) Sankey diagram showing the number of CA1 interneurons (top) and PFC interneurons (bottom) that were classified into different categories. Sizes of partitions are proportional to raw numbers. (**b**) Selectivity index (SI) of all choice-selective interneurons in CA1 (top) and PFC (bottom) on correct trials (left) and incorrect trials (right), aligned to odor-port disengagement. (**c**) Correlation between correct trial SI and incorrect trial SI for CA1 (n = 32, r = –0.80, p = 3.41e-8) and PFC(n = 10, r = -0.38,

*Figure 4 continued on next page*

*Figure 4 continued*

p = 0.278). (**d**) Histograms and waterfall plots of significantly connected PFC-CA1 cell pairs. Peaks falling above zero indicate CA1 leading, whereas peaks falling below zero indicate PFC leading. One-sample Wilcoxson signed rank tests (H0: μ = 0 ms). Top-left: CA1-PFC interneuron pairs, n = 87 pairs, p = 3.5e-8; top-right: CA1 interneuron-PFC pyramidal pairs, n = 283 pairs, p = 2.11e-3; bottom-left: CA1 pyramidal-PFC interneuron pairs, n = 111 pairs, p = 3.4e-3, bottom right: CA1-PFC pyramidal pairs, n = 224 pairs, p = 0.14.

## CA1 and PFC cells phase lock to beta and respiratory rhythms during decision making

We next asked whether there was any relationship between oscillatory phase modulation of neuronal activity and decision accuracy. We observed numerous pyramidal cells and interneurons in CA1 and PFC whose spikes were locked to local rhythms. Spike-phase histograms for example beta-phase and RR-phase locked cells in CA1 and PFC are shown in *Figure 5a–b* (Rayleigh Z test alpha = 0.05, CA1: n=242 spikes, z=15.4, p=1.5e-7; PFC: n=99 spikes, z=40.9%, p=0.007), and the preference for all cells is shown in polar plots (*Figure 5g–h*). A large subset of cells within the population of task-responsive cells in CA1 (*Figure 5c*, CA1 Pyr locked to local RR n=113/138, 82%, interneurons to local RR n=74/83, 89%) and a smaller population of cells in PFC were significantly locked to the local RR (*Figure 5c*, PFC Pyr to local RR n=42/185, 23%, PFC interneurons to local RR n=12/28, 43%). Surprisingly though, while a modest but significant proportion of CA1 pyramidal cells were also coherent to local beta (28/138, 20% binomial test p=4.6e-11) a large fraction of CA1 interneurons were coherent to local beta (36/83, 43%). Conversely, there were very few PFC cells locked to beta overall, as we detected no more cells than chance to be locked to the local beta rhythm (PFC pyr n=8/185 4.3% PFC interneurons n=2/28, or 7%). When we assessed cross-regional spike-beta coherence, we found that there were significantly more CA1 interneurons locked to PFC beta than chance (*Figure 5c–d*, Binomial test on significantly (Bonferonni adjusted alpha = 0.0167) coherent proportion, see **Methods**; CA1 interneurons coherent maximally to PFC beta n=9/83 11%, p=1.7e-20 to OB beta 2/138 1.5% p=0.2), but no high degree of cross regional spike-phase coherence for CA1 Pyramidal cells (CA1 pyr coherent maximally to PFC beta n=11/138, 8.0%, to OB beta 10/138, 7.3%, all p>0.0167) or for PFC cells (*Figure 5c–d*, PFC: pyr maximally coherent to CA1 beta n=10/185, 5.4%, to OB beta n=3/138, 2.2% PFC interneurons maximally coherent to CA1 beta n=3/28 10.7%, to OB beta n=0/28 0%, all p>0.0167). These proportions were higher in the task-responsive population of neurons than in the overall active population (compare *Figure 5c* to *Figure 5—figure supplement 1c*). Overall, CA1 interneurons showed a high degree of both local beta coherence and PFC beta coherence, implicating an important role in cross-regional coordination.

When looking at all task responsive cells regardless of the significance of their spike-phase modulation, phase preference across cells was consistent in CA1 for both rhythms (*Figure 5g–h*, Rayleigh Z-test, CA1: pyr n=138, beta p=5.4e-5, RR *P*=5.9e-4, int n=83 beta p=2.62e-4 RR p=5.61e-6) even though a higher percentage of cells in both regions were phase locked to RR (*Figure 5c* and z-test for proportions, CA1: p=6.8e-23; PFC: p=8.8e-9) and depth of this modulation was much stronger for RR compared to beta (*Figure 5—figure supplement 1a*). However, in PFC only interneurons showed a consistent phase preference to RR (PFC pyr n=185 in n=36, all p>0.05 except for IN to RR p=3.1e-5). The degree of phase-locking for CA1 and PFC neurons is stronger for CA1 or PFC beta than to OB beta (*Figure 5d*, *Figure 5—figure supplement 1b, i, j*), suggesting that this phase-coherent firing is not simply due to common inputs from OB. Furthermore, more cells in CA1 and PFC pyramidal cells were locked to rhythms in CA1 than to those in PFC or in OB (*Figure 5d*). Together with the greater degree of CA1 interneuron coherence to beta across regions, these data suggest a network mechanism wherein CA1 interneurons and CA1 beta play a key role in coordinating CA1-PFC-OB interactions to support odor guided decision making. This beta-driven network is likely different than the RR-driven network, as in both CA1 and PFC, the proportion of cells that were phase locked to both beta *and* RR was no different than chance, given the percentages of cells that were modulated by either rhythm (*Figure 5—figure supplement 1d—e*).

Are cells that are modulated by beta the same population of cells that are selective for the upcoming decision? To test this, we compared the number of pyramidal cells that were both choice-selective *and* phase-locked (within and across regions) to the fraction that would be expected by chance, given the total percentages of cells that are choice-selective or phase-locked. Interestingly, we found that the number of cells that met both criteria was no different than chance for either CA1 or

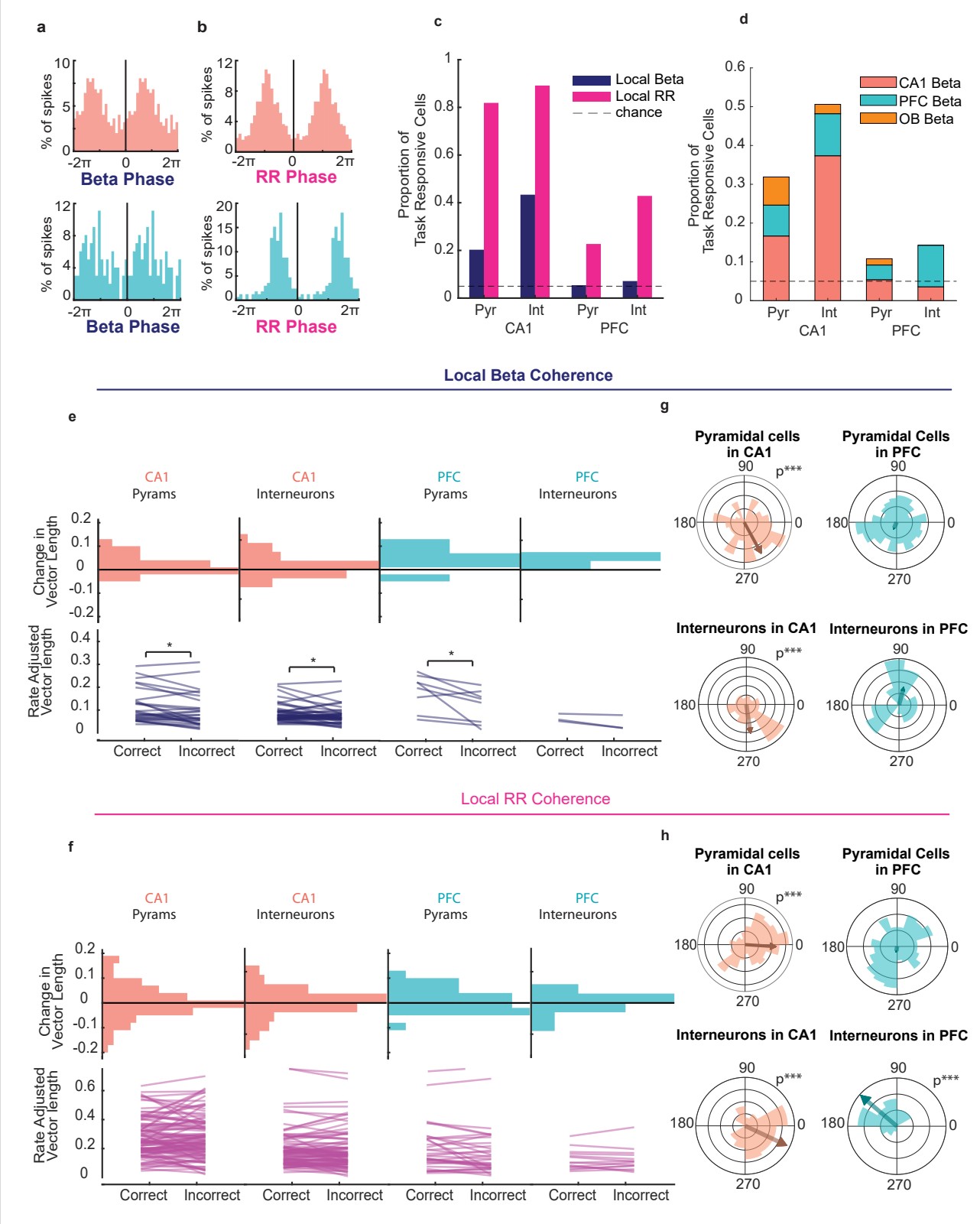

**Figure 5.** CA1 and PFC cells phase lock to beta and respiratory rhythms during decision making. (**a**) Example spike-phase histograms from two example cells that are phase locked to the beta rhythm. Top: CA1 cell (Rayleigh Z test, n = 242 spikes, z = 15.5, p = 1.5e-7); bottom: PFC cell (n = 99 spikes, z = 4.86, p = 0.007). Phase axes are duplicated for visibility. (**b**) Example spike-phase histograms from two example cells that are phase locked to RR. Top: CA1 cell (Rayleigh Z test, n = 870 spikes, z = 135.3, p = 6.3e-62); bottom: PFC cell (n = 172 spikes, z = 73.9, p = 3.9e-37). (**c**) Percentage of task

*Figure 5 continued on next page*

*Figure 5 continued*

responsive cells in PFC and CA1 that were locked to local Beta or RR rhythms (CA1 Pyramidal cells n = 138, INs = 96, PFC Pyramidal cells n = 185, INs n = 36). (**d**) Percentage of task responsive cells in PFC and CA1 locked primarily to Beta in any region. Cells were counted only once and if locked to multiple regions assigned to that which they were locked most significantly(n = same as above). (**e**) Rate Adjusted Vector Lengths (top) and histogram of change in vector length from correct to incorrect trials(bottom) for beta locked task responsive cells in PFC and CA1 that were significantly locked to the local beta rhythm (even if better locked to other regions' LFP) (signed-rank tests, signed-rank tests, CA1: pyr n = 28, p = 0.014, IN n=36, p=0.038; PFC: pyr n = 8, p = 0.039, IN n=2, p>.05). (**f**) Same as e, but for cells that were significantly locked to the Respiratory Rhythm (Signed Rank test CA1 Pyr n=113, n.s, INs n=74, n.s., PFC pyr n=42, n.s. INs n=12, n.s.). (**g**) Polar histogram of preferred beta phases for all task responsive CA1 pyramidal cells and Interneurons (left)(pyr n=138, p=5.4e-5, IN n=96, p=2.6e-4) and PFC pyramidal cells and interneurons (right) (pyr n=185, p=0.648, IN n=36, p=0.50). (**h**) As in (**f**), but for all cells relative to local RR (CA1 pyr n=138, p=5.9e-4, IN n=96, p=5.6e-6. PFC pyr n=185 p=0.79, IN n=36, p=0.3.1e-5). (Asterisks indicate significance; *<0.05, **<0.01, ***<0.001).

The online version of this article includes the following figure supplement(s) for figure 5:

**Figure supplement 1.** Phase locking to beta and respiratory rhythms.

**Figure supplement 2.** Cross-regional beta phase locking for correct vs.incorrect trials.

PFC (*Figure 5—figure supplement 1f*, binomial tests; all p>0.05). We found similar results for interneurons in each region, as well as for RR modulation (*Figure 5—figure supplement 1g–h*). It should be noted that since such a large majority of neurons in CA1 were significantly phase locked to RR, it is unsurprising that many choice-selective cells were phase locked to this rhythm. However, this result indicates that the choice-selective cells were no more or less likely than non-selective cells to phase lock to RR as well. These results together suggest that the cells which are selective for the primary task parameter of odor-cued associative decisions may not be driven directly by the beta rhythm through phase-locking.

We next asked whether the strength of phase locking to the rhythms was still indicative of decision accuracy by comparing correct versus incorrect trials. To do this, we compared each cells mean vector length (MVL) after adjusting for rate differences (*Rangel et al., 2016*), a measure of non-uniformity in the spike-phase distribution, on correct versus incorrect trials for phase-coherent cells. For each cell, we calculated the MVL for the spike-phase distribution on correct trials and incorrect trials separately after correcting for trial count and compared these two paired distributions. We found that both pyramidal cells and interneurons in CA1 and pyramidal cells in PFC that were phase-locked to the local beta rhythm exhibited a lower MVL on incorrect trials (*Figure 5e*, signed-rank tests, CA1: pyr n=28, p=0.014, IN n=36, p=0.038; PFC: pyr n=8, p=0.039, in n=2, these few interneurons did show a decrease). There was no significant effect for local RR phase locking (*Figure 5f*, signed-rank test, no significance found). Interestingly, cross-region spike-phase coherence showed a similar relationship to decision accuracy for CA1 phase-coherent interneurons (*Figure 5—figure supplement 2* n=20, p=0.0023) suggesting a central role for CA1 interneurons in synchronizing PFC-CA1 activity and enabling a correct choice. This suggests that a sub-population of beta modulated cells across PFC and CA1 may play a role in supporting accurate utilization of odor-place associations for making decisions.

## Neural ensemble responses in CA1 and PFC during decision making predict the upcoming choice

We next examined how ensemble dynamics underlie the neural representation of decisions informed by odor-cued recall. We first considered only pyramidal cells that were task-responsive for the ensemble analyses. For this population of task-responsive cells, we found that the distribution of the peak response times in both CA1 and PFC tiled the entire decision time window (*Figure 6a*).

The temporal evolution of CA1 and PFC ensemble responses in individual sessions was quantified to infer the timing of neural discrimination. Principal component analysis (PCA) was applied to visualize the temporal dynamics of population activity during the decision making period, and the Euclidean distance between left-bound and right-bound neural trajectories was used as a measure of neural discrimination (see **Methods**). *Figure 6b* shows an example session with average trajectories of CA1 population activity using the first three PC dimensions for the two choices (n=12 CA1 neurons in ensemble, 100ms bins; n=43 left-bound and n=45 right-bound trials). The two trajectories show a rapid evolution and separation within a few hundred milliseconds after odor onset. The Euclidean distance between left-bound and right-bound average trajectories was compared to a chance-level distance distribution computed by shuffling the trial identities across trials and creating a null distribution. The

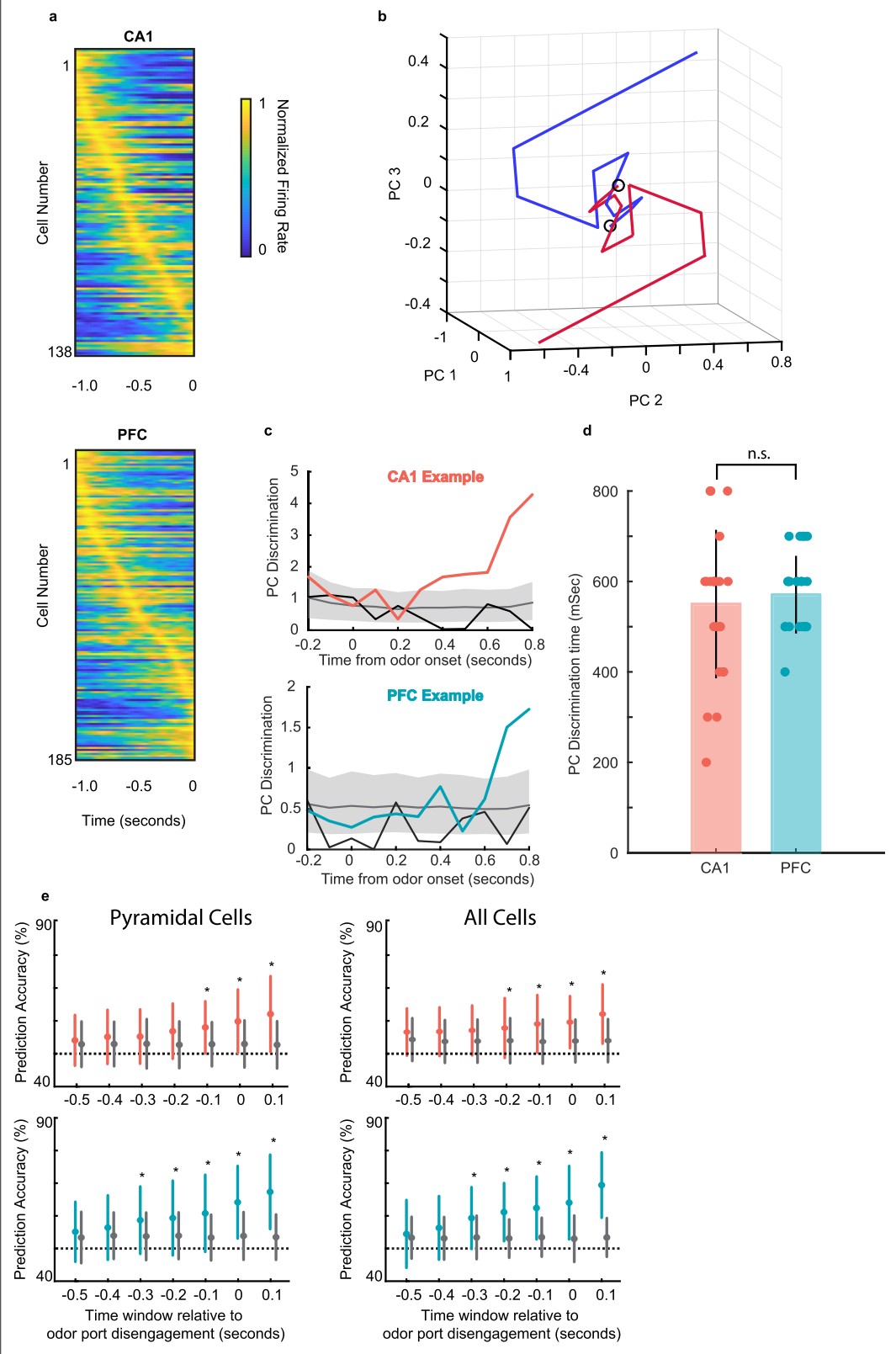

**Figure 6.** Neural ensemble responses during decision making predict the upcoming choice. (**a**) Normalized firing rate of all task-responsive cells in CA1 (top) and PFC (bottom) during a 1 second window before odor port disengagement. (**b**) Example session with average trajectories of CA1 population activity using the first three PC dimensions for the two choices (n = 43, 45 left-bound (blue) and right-bound (red) trials; n = 12 CA1 neurons

*Figure 6 continued on next page*

*Figure 6 continued*

in ensemble, 100 ms bins). The two trajectories showing a rapid evolution and separation within a few hundred ms after odor onset. (**c**) Euclidean distance between left-bound and right-bound average PC trajectories for an example CA1 ensemble (top, n = 12 neurons) and PFC ensemble (bottom, n = 10 neurons) was compared to a chance-level distance distribution computed by shuffling the odor identities across trials and creating a null distribution. Grey line and shading are bootstrap permuted null distribution (mean and 95% CI), and black line is a control distance composed of mean even vs. odd trial PC distance for right and left odors separately. (**d**) Neural discrimination times for CA1 and PFC locked to odor onset (n = 24 sessions CA1 range: 200–800 ms; PFC range, 400–700 ms) were similar (*Figure 7c*; sign rank test, p = 0.68). (**e**) Neural discrimination times during odor sampling aligned to decision time. Colored error bars indicate real data mean ± s.d., whereas gray error bars indicate shuffled data mean ± s.d. Cells used for prediction include all task-responsive putative pyramidal cells in CA1 (top) and PFC (bottom) or all active neurons including interneurons (right). Stars indicate prediction time windows where the fraction of correctly predicted trials was significantly higher than the fraction from the shuffled data (rank-sum tests, * = p < 0.05).

timepoint at which the population responses were considered significantly distinct from each other was defined as the first bin at which the real distance between trajectories surpassed the 95% confidence interval of the null (shuffled) distribution (shown in *Figure 6c* for one example session for CA1 and PFC each). A second even-odd shuffle (*Figure 6c* black lines) failed to cross significance in any session, suggesting the emergence of a robust discriminatory neural code during decision-making. Neural discrimination times for CA1 and PFC (n=24 sessions across 8 animals; threshold of 4 task-responsive cells, CA1: 200–800ms; PFC: 400–700ms) were similar (*Figure 6d*; p=0.68, sign-rank test).

To determine whether the animal's upcoming behavior could indeed be predicted by neural ensembles before the decision was executed, we trained a generalized linear model (GLM) to predict the animal's choice based on ensemble activity during the decision-making period (See **Methods**). When we performed this analysis using task-responsive putative pyramidal cells we found that reward choice could be accurately predicted 0.1 s prior to odor port disengagement by CA1 ensembles, and 0.3 s prior by PFC ensembles (*Figure 6e* **left**). We also performed the same analysis but after included both task-responsive pyramidal and interneurons in the ensembles (*Figure 6e* **right**). In this case, we found an improvement in prediction for CA1 ensembles: reward prediction was now accurate starting earlier, at 0.2 s prior to decision execution. Prediction by PFC ensembles remained the same. To control for the possibility that inclusion of interneurons improved prediction latency for CA1 simply due to the larger number of cells in the training set, we performed this analysis again by resampling the pyramidal cells to match the total number of pyramidal cells plus interneurons. In this control analysis, we found that the choice could again only be predicted at 0.1 s prior to the decision execution, the same as what we observed with the original sample of pyramidal cells. These results confirm that the animals utilize the recalled odor-place association and make a spatial choice during this odor-sampling period, which is reflected in the activity of task-responsive neural ensembles.

## Representations of choice and space are maintained independently during stem running

Finally, we investigated whether there was any relationship between the activity of CA1 and PFC ensembles during the odor-cued decision-making period and their spatial activity on the maze (excluding the odor sampling and reward periods) as the animals ran through the common central arm on the T-maze on the central and side arms toward reward. We examined units only from sessions in which animals traversed a long T-maze track (see **Methods**, 26 sessions from 6 rats) in order to assess spatial firing characteristics. A large fraction of cells in CA1 and PFC exhibited spatial activity on the maze (see **Methods** for details on spatial parameters, n=344 out of 585 CA1 cells; n=159 out of 288 PFC cells had fields on the track), including units that were both active during the odor period (choice selective and non-selective) and inactive during these periods (*Figure 7a and b*; examples of CA1 and PFC choice-selective cells with spatial fields in *Figure 7a*, examples of CA1 and PFC odor active but non-selective cells with spatial fields shown in *Figure 7—figure supplement 1a*).

Cells that were odor period active were more likely to have spatial fields on the track (odor sampling periods were excluded in spatial responses, see **Methods**), compared to cells that were not (*Figure 7b*, binomial test, CA1: p=0.029; PFC: p=2.7e-5). Additionally, for both CA1 and PFC neurons, we found odor period active cells had higher firing rates overall (*Figure 7—figure supplement 1b*), suggesting

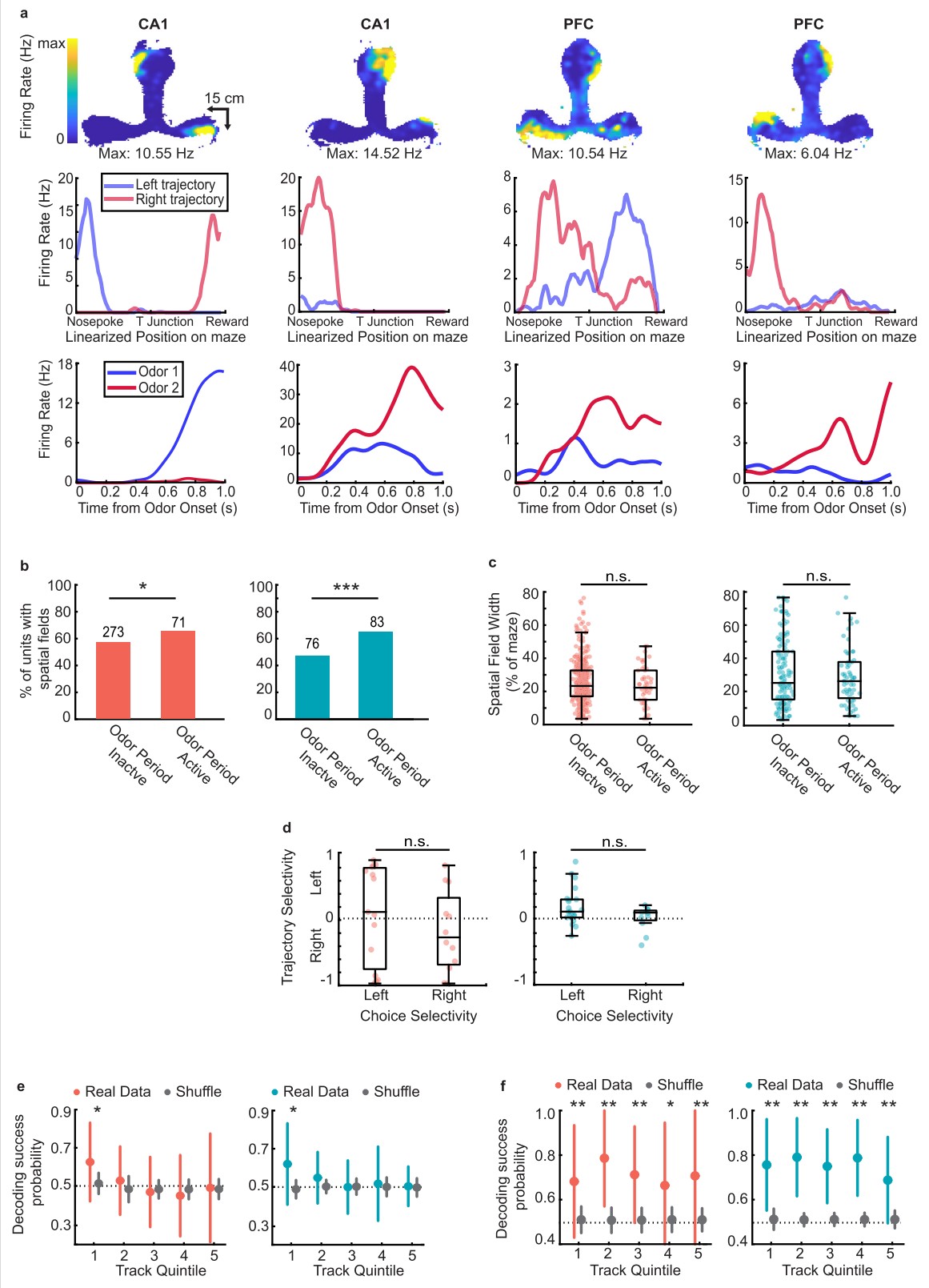

**Figure 7.** Representations of choice and space are maintained independently during delay period. (**a**) Example choice-selective units (CA1, first two columns, PFC, last two columns) with spatial fields on the track. Top row: Heat map illustrating spatial fields during run bouts. Middle row: Linearized spatial tuning curves for outbound left and right run trajectories. Bottom row: PSTHs showing odor responses during decision-making period (Odor selectivity indices, left to right: 0.92,–0.43, −0.40,–0.57). (**b**) Spatial field prevalence in odor period inactive and odor period active cells in CA1 (left) and

*Figure 7 continued on next page*

*Figure 7 continued*

PFC (right). CA1: 273/477 (57%) of inactive cells, and 71/108 (66%) of active cells had spatial fields (binomial test: p = 0.03*); PFC: 76/160 (48%) of inactive cells, and 83/128 (66%) of active cells had spatial fields (binomial test: p = 2.7e-5***). (**c**) Spatial field width for odor period inactive and odor period active units in CA1 (left) and PFC (right). If a unit had a spatial field on both outbound trajectories, each field was counted separately. (Rank-sum tests; CA1: n = 452 fields, p = 0.15; PFC: n = 207 fields, p = 0.92). (**d**) Trajectory selectivity scores of choice-selective units according to preferred choice (Rank sum tests, CA1: p = 0.26; PFC: p = 0.20). Boxplots represent interquartile range and stems represent 95% percentile bounds. (**e**) Decoding of choice identity by naïve Bayesian classifier according to CA1 (left) and PFC (right) task-responsive ensemble activity at five equally sized spatial quintiles along the full run trajectory of the maze. 1 spatial quintile = 24.6 cm (123 cm / 5). Colored error bars indicate real data mean ± s.d. whereas grey error bars indicate shuffled data mean ± s.d. Stars indicate quintiles where the fraction of correctly decoded trials was significantly higher than the fraction from the shuffled data (rank-sum tests, * = p < 0.05). (**f**) As in (**e**), but decoding of choice identity according to CA1 (left) and PFC (right) spatially-modulated ensemble activity at each spatial quintile (rank-sum tests, * = p < 0.05, ** = p < 0.01).

The online version of this article includes the following figure supplement(s) for figure 7:

**Figure supplement 1.** Spatial activity of task-responsive and task-unresponsive neurons.

a possible relationship between cell activity during decision making and maintenance of the decision during the central stem run. To examine this question further, we asked whether the spatial fields in either CA1 or PFC exhibited different characteristics based on whether or not they were active during the decision-making period. Surprisingly however, we found no difference in field width (*Figure 7c*, rank-sum tests, CA1: p=0.15; PFC: p=0.92), field peak rate (*Figure 7—figure supplement 1c*), or field sparsity (*Figure 7—figure supplement 1d*) between cells that were odor period active or inactive in either region.

Further, we also examined trajectory selectivity as animals ran through the central arm and on the side arms toward reward (examples of trajectory selective firing shown in *Figure 7a*, second row). Trajectory selectivity was defined by comparing spatial tuning curves of cells on right versus left trajectories and calculating a selectivity index analogous to the choice selectivity index during the decision-making period (see **Methods**). Interestingly, although trajectory-selective cells were significantly more likely than trajectory-nonselective cells to respond in the decision-making period (*Figure 7—figure supplement 1e*), there was no relationship between the preferred trajectory during run periods and preferred choice during the odor-cued decision-making period (*Figure 7d*, rank sum test CA1: p=0.26; PFC: p=0.20). Thus, although CA1 and PFC cells that were active during odor sampling had higher firing rates and therefore higher engagement while traversing trajectories on the maze, there was no clear relationship between choice selectivity and trajectory selectivity in either CA1 or PFC.

To confirm this result, we sought to determine whether choice selective neuronal activity in the population during decision making persisted into the following run period. To examine this, the linearized T-maze was divided into five equally spaced quintiles and a Bayesian Classifier trained on odor period activity was used to decode the choice identity from the ensemble activity during run at each spatial quintile (see **Methods**; similar results were seen for spatial quartiles). We found that choice identity could be decoded at the first quintile of track, nearest to the odor port, but decoding accuracy diminished to chance level thereafter (*Figure 7e*). In contrast, the animal's upcoming behavioral choice could be accurately decoded at all periods along the trajectory if instead the spatially active ensembles in each respective spatial quintile was used as the training set (*Figure 7f*). This suggests that the choice selectivity that emerges in ensembles during odor sampling (*Figure 6*) persists transiently for a short period past the decision point, but that separate ensembles maintain choice-related information as animals traverse the spatial trajectory, possibly corresponding to theta oscillations during run (*Figure 2—figure supplement 1e*). This is further supported by the fact that, at least for CA1, choice-selective ensemble firing rate is highest within 20 cm of the odor port, and decreases thereafter (*Figure 7—figure supplement 1f*). These results together suggest that the choice-selective ensembles reflecting decision making during the odor-cue period transiently encode the choice during the initial track period, but the decision is subsequently maintained by a distinct ensemble during the working-memory period associated with running on the central arm.

## Discussion

These results provide new insight into the mechanisms of rhythmic coordination of hippocampal–prefrontal ensembles by beta rhythms for odor-cued associations and decision making. We found

that during an odor-place associative memory task, beta and RR rhythms govern physiological coordination between the olfactory bulb, hippocampus, and PFC (OB-CA1--PFC). Crucially, CA1-PFC beta coordination during the decision-making period of this task is linked to decision accuracy. During the odor sampling and decision-making period, task-responsive single unit and ensemble activity in the CA1-PFC network discriminates between choices and can reliably predict upcoming behavior. CA1-PFC-OB beta coherence was elevated during the decision-making period, but we did not find evidence for difference in coherence for correct vs. incorrect trials. Instead, we found that a CA1-PFC sub-population of phase-locked cells exhibited stronger phase modulation by beta rhythms on correct vs. incorrect trials. In particular, CA1 pyramidal cells and interneurons, as well as pyramidal cells in PFC exhibited stronger phase modulation by local beta rhythms during accurate memory-guided decision making. CA1 interneurons exhibited cross-regional spike-phase beta coherence and higher phase modulation to PFC beta during correct trials, and showed temporal coordination with PFC neurons in cross-correlations. This suggests that coordination by odor-driven beta rhythms can sculpt network activity through interneuron modulation, enabling emergence of task-related ensemble dynamics in local circuits to support decision making. Different cell ensembles were engaged in CA1-PFC during decision-making vs. decision execution on the maze. Our results thus suggest that oscillatory modulation across and within CA1, PFC, and OB networks plays a key role in temporal evolution of ensemble dynamics for odor-cued decision making.

The mechanisms underlying encoding and recall of associations between sensory cues in the environment, and subsequent use of these associations to guide decisions, are of fundamental interest. Sensory cue-elicited recall and decision making during goal-directed behavior involves widespread networks encompassing sensory regions, medial temporal memory regions, and prefrontal executive function regions. Prominent odor-driven oscillations have been described in these areas (*Frederick et al., 2016*; *Igarashi et al., 2014*; *Kay and Beshel, 2010*; *Kepecs et al., 2006*; *Lockmann et al., 2016*; *Nguyen Chi et al., 2016*; *Rangel et al., 2016*; *Stopfer et al., 2003*; *Tort et al., 2009*), which can potentially coordinate these long-range networks to enable utilization of familiar olfactory cues to guide behavior. Here, the use of an odor-cued T-maze task allowed us to examine which rhythms enable coordination of olfactory-hippocampal-prefrontal networks, and whether these rhythms play a role in patterning ensemble activity in hippocampal-prefrontal network to enable memory-guided decision making.

In this task, the time period between odor onset and odor port disengagement provided a temporal window corresponding to odor-cued recall and priming of the subsequent decision to turn toward the reward location. We found prevalence of both beta (~20–30 Hz) and RR (~7–8 Hz) across the olfactory-hippocampal-prefrontal network during this odor sampling and decision-making period. There remained a consistent beta-phase and RR phase offset between PFC and CA1 during odor sampling, suggesting a subtler interaction between regions at those frequencies. The strength of beta coherence between the hippocampus and PFC was enhanced during the decision-making period when compared to pre-odor periods as well as compared to the period of immobility during reward consumption. Further, while CA1-PFC RR coherence remained unchanged between odor-cued trials and trials where only air was presented, high beta coherence was specific to odor-cued trials corresponding to a decision-making task and was lower on air-cued trials corresponding to random choices. Based on these findings, we speculate that encountering a familiar odor stimulus and efficiently utilizing this associative memory for a decision, rather than respiration or movement preparation (*Hermer-Vazquez et al., 2007*), elicits the engagement of the hippocampal-prefrontal network by the beta rhythm.

Interestingly, although hippocampal SWRs have been linked to internal recall and planning (*Buzsáki, 2015*; *Carr et al., 2011*; *Joo and Frank, 2018*), there was very low prevalence of hippocampal SWRs during odor sampling, suggesting SWRs are not directly involved in odor-cued decision making while the animal is re-encountering the familiar cue. Instead, our results indicate that beta rhythms play a key role in mediating this process, in addition to coordinating entorhinal-hippocampal networks for odor-cued recall, as described previously (*Igarashi et al., 2014*). In our experiments, rats were very familiar with the task and odor-place associations, with high-performance levels. Therefore, the lack of SWRs during the decision-making period could reflect that these memories had been previously consolidated into neocortex during learning, resulting in a shift from largely hippocampal-dependent processing (*Igarashi et al., 2014*) to processing that is more reliant on hippocampal-cortical dialogue via the beta rhythm.

Odor-driven gamma rhythms at 40–100 Hz have also been reported in odor-memory tasks (***Beshel et al., 2007***; ***Frederick et al., 2016***; ***Kay, 2014***), and may play a complementary role with beta oscillations. It has been hypothesized that local olfactory networks may be governed by gamma rhythms for odor processing early in stimulus sampling, before shifting to a beta-dominant network state as downstream regions are engaged (***Frederick et al., 2016***; ***Martin et al., 2006***). Our results corroborate this hypothesis by reinforcing the role of the beta rhythm in long-range coordination with a wider network outside the olfactory system for cognitive processing of odor stimuli. We speculate that sensory modality-specific rhythms may play a general role in cue-driven mnemonic decision making by coordinating sensory and cognitive areas, which can be tested in future studies.

Our analysis of single units in the hippocampus and prefrontal cortex revealed a group of cells that were selectively responsive in the odor task period, similar to previous reports in the hippocampus (***Eichenbaum et al., 1987***; ***Igarashi et al., 2014***; ***Terada et al., 2017***). However, we found that these selective cells often exhibited 'opposite' selectivity on incorrect trials compared to correct trials, indicating that these cells do not respond solely to the perceptual qualities of the odor, in which case the response would be similar on all presentations of a particular odor regardless of the trial outcome. Instead, this response suggests coding of the behavioral choice associated with the odor cue, resembling findings from studies of combined cue modalities and cue-context associations (***Allen et al., 2016***; ***Ferbinteanu et al., 2011***; ***Fitzgerald et al., 2011***; ***Komorowski et al., 2009***; ***McKenzie et al., 2014***; ***Otto and Eichenbaum, 1992b***; ***Schoenbaum and Eichenbaum, 1995b***; ***Terada et al., 2017***). Thus, the CA1-PFC neural response during odor sampling corresponds to the decision in response to the odor-place association. Future studies that combine recording from other olfactory regions, such as piriform cortex, together with hippocampus and PFC can potentially unveil how the cue representation evolves from odor perception to the recall-based decision.

Task-responsive neural populations exhibited differential activity patterns in response to the two odor-associations during decision making, and the animal's upcoming choice could be predicted from this ensemble activity. Overall, the timing of ensemble odor discrimination preceded choice execution, indicating that the emergence of the neuronal representation can prime the cognitive decision. This emergence of neural ensembles that reflect associative memories governing decision making may also be indicative of a physiological mechanism for dynamic reactivation of memory engrams upon re-encountering the familiar odor stimuli (***DeNardo et al., 2019***; ***Liu et al., 2012***). It remains unclear, however, how these neural dynamics are modulated by beta oscillations and how they evolve as animals learn new associations. Our data lacked large, simultaneously recorded task-responsive ensembles, which precluded analyses investigating the link between ensemble dynamics and beta coherence.

Similarly, CA1-OB beta coherence is strong during odor memory tasks and is known to increase over the course of learning (***Martin et al., 2007***). Although CA1-OB beta coherence was high during odor sampling in our task, we observed no significant relationship between CA1-OB coherence and task performance. Since previous reports suggest that this coherence is linked to the learning process, it is possible that a more direct relationship between coherence strength and performance would be observed only during learning, but not in this case where the odor association and task are familiar. Choice-coding ensemble dynamics may evolve in parallel with the increase in CA1-OB beta coherence over learning, similar to what has been reported in the hippocampal-entorhinal circuit (***Igarashi et al., 2014***). This possibility can be investigated by examining network dynamics and ensemble activity across multiple olfactory and cognitive areas during novel odor-place learning or reversal learning.

It is also noteworthy that we observed different mechanisms underlying odor-cued decision making versus subsequent maintenance of the representation of the choice on the central stem. Odor-driven beta coherence and phase modulation has a role in accurate decisions, and possibly leads to emergence of ensemble selectivity which is predictive of the animal's upcoming choices. However, these ensembles maintain their choice-selectivity only transiently after the decision execution as animals embark on the spatial trajectory toward reward. This selective coding is not maintained by the same ensembles during running, but instead other spatially modulated cells in CA1 and PFC maintain choice coding on the track, including the central stem run period activity. Thus, beta driven ensembles mediate memory-guided decision making in the hippocampal-prefrontal network, but choice selective activity is then subsequently maintained by theta-driven spatially modulated activity, similar to reports in spatial working memory tasks (***Taxidis et al., 2020***).

The temporal evolution of ensemble activity underlying odor-cued decision making is reminiscent of temporal coding of odor stimuli mediated by oscillations in the olfactory system (*Kepecs et al., 2006*; *Stopfer et al., 2003*). Previous findings indicate that spike timing modulation according to the phase of prominent network rhythms can organize task-encoding neural ensembles to support memory guided behavior and decision-making (*Benchenane et al., 2010*; *Buschman et al., 2012*; *Papale et al., 2016*; *Terada et al., 2017*; *Wikenheiser and Redish, 2015*; *Zielinski et al., 2019*). Interestingly however, we found that although the strength of local beta phase-modulation was linked to correct decisions, putative neurons that encoded the choice were no more likely to be phase-locked than chance level. Further, we found evidence that different populations of cells are modulated by the two ongoing rhythms during decision making, implying that beta and RR represent two simultaneous modes of rhythmic coordination in the network (*Rangel et al., 2016*). This joint modulation of the underlying cell populations, along with our finding of strong cross-frequency coupling between the two rhythms, leads us to the interpretation that beta and RR coordinate the network cooperatively during odor-cued decision making. Previous reports of RR in olfactory processing (*Karalis and Sirota, 2022*; *Kay, 2005*; *Kepecs et al., 2006*; *Nguyen Chi et al., 2016*), along with evidence that multiple rhythms can simultaneously modulate ongoing processes (*Lisman and Jensen, 2013*; *Rangel et al., 2016*; *Zhong et al., 2017*), suggests that RR could be the dominant modulator coordinating the sensory element of the task, while beta coordination is key for employing the sensory cued association to make a decision. This interpretation is further supported by our results from un-cued air sessions, which did not involve memory-guided behavior, in which we found a reduction in beta coherence in the network but no change in RR coherence. This suggests that the sensory component of this process is maintained as the animal continues sniffing even in the absence of an explicit cue, whereas beta coordination is only engaged when a familiar cue is utilized for a decision.

The stronger local beta phase modulation of spiking activity during correct trials strongly suggests that beta coordination is related to task performance. We also found that a large population of CA1 interneurons were strongly phase-modulated by the local CA1 beta rhythm as well as cross-regional PFC beta, and further that the strength of this modulation was linked to task performance. The higher beta phase modulation of CA1 interneurons also suggests that entrainment of interneurons to the network-wide beta rhythm is linked to task performance. PFC interneurons also showed temporal coordination with CA1 interneurons and pyramidal cells, and the strong skew towards PFC interneuron's leading CA1 interneurons and CA1 pyramidal cells suggests top down influence of the PFC on CA1. Since interneurons have a pre-dominant effect on local network activity, rhythmic modulation of interneurons, possibly through a common input to the CA1-PFC network, can potentially play a key role in temporally organizing ensemble responses to enable processing of odor-cued associations for translation to decisions. (*Andrianova et al., 2021*; *Dolleman-van der Weel et al., 2019*; *Schlecht et al., 2022*; *Varela et al., 2014*). Beta oscillations may thus lay a role in establishing communication and organization of activity in sensory and cognitive networks enabling decisions based on cued associations. This oscillatory phase-modulation may be indicative of a general network state that enables coordination.

## Methods

**Key resources table**

| Reagent type (species) or resource | Designation | Source or reference | Identifiers | Additional information |
|---|---|---|---|---|
| Strain, strain background (Rat) | Long Evans | Charles River | Cat#: Crl:LE 006; RRID: RGD_2308852 | |
| Chemical compound, drug | Ethyl Butyrate | Sigma-Aldrich | Cat#: H2805 | |
| Chemical compound, drug | Heptanol | Sigma-Aldrich | Cat#: E15701 | |
| Chemical compound, drug | Formaldehyde | Thermo Fisher Scientific | Cat#: 50-00-0,67561,7732-18-5 | |
| Software, algorithm | Matlab | Mathworks, MA | RRID: SCR_001622 | |

*Continued on next page*

*Continued*

| Reagent type (species) or resource | Designation | Source or reference | Identifiers | Additional information |
|---|---|---|---|---|
| Software, algorithm | Mountainsort | *Barnett et al., 2016*; *Chung et al., 2017* | https://github.com/magland/mountainlab; RRID: SCR_017446 | |
| Software, algorithm | Chronux | Partha Mitra | http://www.chronux.org/ | |
| Other | 128-channel data acquisition system | SpikeGadgets | http://www.spikegadgets.com | Electrophysiology data acuisition system |
| Other | Olfactometer | MedAssociates Inc | Cat#: PHM-275 | Two Channel Dilution Olfactometer |
| Other | 12.7 μm NiCr tetrode wire | Sandvik | Cat#: PX000004 | Insulated wire to make electrodes |

## Animals

All experimental procedures were approved by the Brandeis University Institutional Animal Care and Usage Committee (IACUC) and conformed to US National Institutes of Health guidelines. Eight male Long-Evans rats (3–6 months, 450–650 g, RRID: RGD_2308852) were used for experiments. Animals were housed individually in a dedicated climate-controlled animal facility on a 12 hr light/dark cycle. All experiments were carried out during light cycle. Upon arrival, animals were provided ad libitum access to food and water and handled regularly to habituate them to human contact.

## Behavior apparatus

An olfactometer (MedAssociates Inc) was used for dispensing odors. The olfactometer continuously dispensed clean air to the odor port until receiving a signal to open a solenoid valve which caused air to flow through liquid odorants, resulting in odorized air dispensed to the odor port. A vacuum tube attached to the odor port was used to continuously collect any residual odorized air between trials. Infrared beams were used at the odor port and reward wells to determine the precise timing of entry and exit from these areas.

## Odor-place association task training

Once rats reached a minimum threshold weight of 450 g, they were food restricted to no less than 85% of their free-feeding baseline weight. For initial behavioral training, rats were familiarized with the behavior room, the sleep box, running on a raised track to receive evaporated milk reward, and sniffing odors presented in the odor port. Following this habituation, rats were trained to hold their nose in the odor port for a minimum of 500ms, with an auditory tone indicating when this 500 ms time threshold was reached. However, the rats could continue to sniff the odor for any longer duration of time, and odor would be continuously dispensed until they disengaged from the odor port. Throughout all training and experiments, if the rat exited the odor port before this threshold was reached, no reward was dispensed regardless of the rat's choice and the rat was required to re-initiate the trial. These prematurely terminated trials were excluded from all analyses. For each trial, the odor sampling period was defined as the time from the onset when rat's nose broke the infrared beam to offset when the beam break was terminated, and odor stimulus was stopped.

Animals were subsequently trained on the olfactory association. On each trial, one of two possible odors was dispensed at the odor port in a pseudo-random order. Odor 1 (Heptanol – pine/citrus scent) indicated that the rat should go to Reward 1 to receive reward. Odor 2 (ethyl butyrate – strawberry scent) indicated that the rat should choose Reward 2 (*Figure 1a*). If the rat ultimately made the correct choice, evaporated milk reward would be dispensed at the chosen reward port upon triggering the infrared beam, whereas upon an incorrect choice no reward was dispensed.

Associative trace memory training was shaped in steps. The rats first learned the association between the odors and 'right' vs 'left' reward locations with reward wells close to the odor port. On subsequent training days, the reward wells were moved further and further away from the odor port, until rats could comfortably perform the task on the full T-maze (81 cm long center stem (spatial

delay), and 43 cm long reward arms). Three were not trained to run on the full T-maze - two of these required to perform the association with the reward wells on either side of the odor port while the last animal ran with a shortened stem (40 cm instead of 81 cm), but the reward wells were still at the ends of the T arms (truncated mazes). Task performance was calculated as the proportion of correct trials. Rats were trained until they could perform the task with at least 80% accuracy for 3 consecutive days.

Once training was complete, animals were once again provided with ad libitum access to food until they reached at least 600 g, before undergoing surgery. After surgery but before recording, rats were briefly re-trained on the association until they could again perform the task with at least 80% accuracy. On recording days, animals were allowed to continue performing the task until they reached satiation, about 100–150 trials per day. Task epochs were interleaved by sleep epochs, in which rats spent about 20 min in an opaque sleep box, with a sleep epoch as the first and last epoch of each day.

### Un-cued air sessions

Four of the animals were tested on one session each in which for every trial clean air was dispensed at the odor port, instead of two distinct odors. This session was always between two above criterion performing sessions to reduce the likelihood of the rat not sniffing or not knowing the task rules. Reward was available at a randomly chosen reward port on each trial; although, since the trials were un-cued, the animals would often randomly choose the 'incorrect' side and no reward was dispensed (*Figure 1—figure supplement 1f*). All other aspects of the task were the same as the odor-cued task.

### Surgical procedures

Surgical implantation techniques were performed as described previously (*Shin et al., 2019*; *Tang et al., 2017*; *Tang et al., 2021*). Briefly, animals were implanted with a microdrive array containing 30 independently moveable tetrodes (12.7 µm NiCr tetrode wire) targeting right dorsal hippocampal region CA1 (–3.6 mm AP and + 2.2 mm ML, 10–12 tetrodes), right PFC (+3.0 mm AP and + 0.7 mm ML10-13 tetrodes), and right olfactory bulb (+7.2 mm AP and + 0.8 mm ML, 1–4 tetrodes). Post-operative analgesic care was provided for 48 hr after surgery to minimize pain and discomfort. In two of the eight animals used for experiments, a nasal thermocouple was implanted in addition to the microdrive array to record the respiratory rhythm. The thermocouple was placed in the left nostril though a hole drilled in the skull at + 7.5 mm anterior to the cribriform suture. The thermocouple wire was secured using dental acrylic and soldered to the same printed circuit board as the tetrodes.

### Tetrode recordings

For 1–2 weeks following surgery, tetrodes were gradually lowered to the desired depths. Hippocampal tetrodes were targeted to the pyramidal layer of CA1 using characteristic EEG patterns (sharp wave polarity, theta modulation) and neural firing patterns as previously described (*Jadhav et al., 2012*; *Jadhav et al., 2016*). The final placement of tetrodes was confirmed in histological preparations using Nissl staining post-mortem. One tetrode in corpus callosum served as hippocampal reference, and another tetrode in overlying cortical regions with no spiking signal served as PFC reference. A ground screw (GND) installed in the skull overlying cerebellum also served as a reference, and LFP was recorded relative to this GND. All spiking activity was recorded relative to the local reference tetrode. Only LFP activity was recorded from the olfactory bulb. Electrodes were not moved at least 4 hr before and during the recording day to reduce drift, and were micro-adjusted at the end of each recording day to sample new cell populations.

Data were collected using a SpikeGadgets 128-channel data acquisition system and software (SpikeGadgets LLC). Spike data were sampled at 30 kHz and bandpass filtered between 600 Hz and 6 kHz. LFP signals were sampled at 1.5 kHz and bandpass filtered between 0.5 Hz and 400 Hz. The animal's position and running speed were recorded with an overhead color CCD camera (30 fps) and tracked by color LEDs affixed to the headstage.

### Data analysis and statistics

All data analysis was performed in MATLAB using custom code unless otherwise noted. Error bars indicate standard deviation unless otherwise noted. Significance was defined using an alpha of 0.05. Statistical details including tests used, p-values, and n values can be found in figure legends, and are described in-depth below.

## Local field potential

Trial-averaged spectrograms and coherograms were calculated using multi-taper spectral methods included in the Chronux package for MATLAB (http://www.chronux.org/). To get the beta filtered local field potential (LFP) signal, raw LFP (with respect to GND) was band pass filtered at 20–30 Hz using a zero-phase IIR filter. Amplitude, phase, and envelope magnitude of the signals were obtained using a Hilbert transform (for each rhythm zero phase was defined as the peak of the sine wave). Beta power and coherence were z-scored to the epoch mean. Similarly, the respiratory rhythm signal was obtained by band pass filtering the raw LFP at 7–8 Hz. All comparisons between conditions (pre-post odor period, correct vs incorrect, and odor vs air or reward period) were calculated as within-session difference in coherence or power, and the mean and SEM of those difference values across sessions were plotted.

## Bootstrap tests

Bootstrap tests for *Figure 2g* were performed when coherence on odor-cued and air-cued trials were compared, to account for the much larger percentage of odor-cued trials. Data points on odor trials were randomly down-sampled with replacement to match the number of air trials, and the mean was calculated for this new set of datapoints. This down-sampling was done 1000 times, to create a new, bootstrapped distribution of means. The observed mean value for the air trial distribution was compared to the bootstrapped distribution. p-Values were calculated by counting the number of values in the bootstrapped distribution that were greater than or equal to the observed mean value, and dividing by the total number of re-samples (1000). p-Values less than 0.05 were considered statistically significant, and thus rejected the null hypothesis that the two observed distributions had equal means.

## Sharp-wave ripple detection

Hippocampal sharp-wave ripples were detected as previously described (*Jadhav et al., 2016*; *Shin et al., 2019*; *Tang et al., 2017*). Briefly, the locally referenced LFP signal from CA1 tetrodes was filtered in the ripple band (150–250 Hz), and the envelope of the ripple-filtered LFPs was determined using a Hilbert transform. SWR events were detected as contiguous periods when the envelope stayed above 3 SD of the mean on at least one tetrode for at least 15ms.

## Cross-frequency coupling

Phase-amplitude coupling between RR and beta was computed for *Figure 2—figure supplement 1f* as previously described (*Tort et al., 2010*). In brief, the phases of RR were divided into 20 degree bins, and the mean amplitude of the beta rhythm at each phase was calculated. The mean amplitudes were then normalized by dividing each bin by the sum of amplitudes across all bins. The strength of phase-amplitude coupling was determined by comparing the amplitude distribution to a uniform distribution by calculating the modulation index (MI), which is based on the Kullback–Leibler (KL) distance but normalized so that values fall between 0 and 1, where a value of 0 indicates a uniform distribution of beta amplitudes across RR phases. Significance was determined by comparing the calculated MI to a null distribution generated by shuffling the trial number assignments of the RR phase series; this maintains the structure of the underlying rhythm but randomly aligns the beta amplitudes on each trial to each trial sequence RR phases.

## Single unit analysis

Spike sorting was done semi-automatically using MountainSort (*Barnett et al., 2016*; *Chung et al., 2017*), with manual curation. Only well-isolated units were used for analysis. Putative interneurons and pyramidal cells were classified based on average firing rate and spike width, as described previously (*Shin et al., 2019*; *Tang et al., 2017*). Units were classified as interneurons if they had an average firing rate exceeding 7 Hz and an average spike width under 0.3ms. All other units were considered putative pyramidal cells. Units were excluded from analysis if they had fewer than 100 spikes across all task epochs. A number of pyramidal cells only had spikes during sleep epochs, and these were excluded from analysis (*Jadhav et al., 2016*; *Karlsson and Frank, 2009*). For analyses during the odor-sampling and decision-making period (*Figures 3–6*), cells were included with the criterion of spikes at least

equal to the number of trials (*Figure 3g–h*, 'Odor Period Active' and 'Inactive Cells'), since all analysis done here focused exclusively on the task periods.

## Task responsiveness

Task responsiveness was calculated as a change in firing rate following odor onset, compared to a pre-stimulus period of the same length of time as the odor sampling period on each trial leading up to odor port engagement. Sampling events were only considered if the animal held its nose in the odor port for longer than 0.50 s and proceeded to a reward port. Statistical significance was determined from the Wilcoxon signed rank test of those trial-matched rate differences for each odor identity separately. If a cell showed a significant change up or down from baseline for at least one of the odors, it was considered task responsive. Of note, there was only a single unit analyzed whose firing rate changed in opposite directions from baseline for the two odors, the remainder showed the same direction of change in firing rate for the two odors.

## Choice selectivity

Firing rates during odor sampling were calculated as the number of spikes as a function of time from odor-port engagement to odor-port disengagement. Choice selectivity was calculated using the following equation:

$$SI = \frac{\lambda_1 - \lambda_2}{\lambda_1 + \lambda_2}$$

Where $\lambda_1$ is the firing rate vector for Odor 1 trials, and $\lambda_2$ is firing rate vector for Odor 2 trials. SI = 1 indicates that the cell only responded on Odor 1 trials, whereas *SI* = –1 indicates that the cell only responded on Odor 2 trials. To determine significance, a null distribution was generated in which the odor identities were shuffled across trials. Cells were considered choice-selective if the SI fell outside of 1.5 s.d. from the mean of the null distribution.

## Corrected spike cross-correlogram

To correct for the triangular shape in the cross-correlogram, we utilized a spike-shuffling procedure similar to that previously described (*Kay et al., 2020*). In brief, all spikes from one cell were jittered randomly +/- 50 ms and the cross correlogram was recalculated from –0.15 seconds to.15 s in 2.5 ms bins. This procedure was performed 1000 times, and the pointwise z-score of the real cross-correlogram was calculated from the shuffle. The highest peak within +/- 100 ms that achieved significance (p<0.05), was chosen as the significant peak in the ccg, and only those cross correlograms in which there was a significant peak were displayed (and smoothed with a Gaussian kernel with σ=1 bin).

## Phase locking

For all phase-locking analyses, the tetrode in each region with the most cells on each given day was used to measure oscillatory phase of either beta or RR. Phase locking of individual cells to the beta and respiratory rhythms was calculated by pooling the spike phases of each cell during the odor-sampling periods and performing a Rayleigh Z test for circular non-uniformity. When comparing phase coherence during correct trials versus that during incorrect trials, we adopted a down sampling strategy to adjust for rate differences, as described previously (*Rangel et al., 2016*). First, we matched the number of spikes during correct trials to that during incorrect trials. We then bootstrapped a mean vector length for the downsampled correct trials 1000 times and directly compared that to the MVL during incorrect trials. For calculation of the mean phase preference of each cell, we calculated the mean phase of all spikes during correct trials assuming a Von-Mises distribution.

## Principal component analysis (PCA)

We used PCA to visualize population activity of CA1 and PFC ensembles over time individual sessions. Spiking activity in the decision-making period aligned to odor onset was binned (binsize = 100ms, window = 0–1 s), and a firing rate matrix was constructed where each row represents a bin and each column represents a neuron. Only sessions with at least four task-responsive neurons (threshold applied for both CA1 and PFC; CA1 and PFC ensembles were examined separately) were used for analyses. We used PCA to find the principal component coefficients of the matrix and applied the

coefficients to the population activity for left-bound vs. right-bound trials. Population activity was projected onto the PC space. The first 3 PCs were used for visualization and analysis. The Euclidean distance between left-bound and right-bound average trajectories was compared to a chance-level distance distribution computed by shuffling the trial identities across trials and creating a null distribution. The timepoint at which the population responses were considered significantly distinct from each other was defined as the first timepoint at which the real distance between trajectories surpassed the 95% confidence interval of the null (shuffled) distribution. Similar results were obtained for Euclidean distance computed with the first 3 PCs and for Euclidean distance computed for all neurons without dimensionality reduction.

## GLM

A generalized linear model (GLM) with a log link function was constructed to predict reward choice based on neural activity during the odor sampling period. Activity from all task-responsive neurons that were active on at least 10 trials was included. Neural population activity from different length time bins from 0.1 s to 1 s aligned to odor onset was used for prediction, that is 0–0.1 s, 0–0.2 s … 0–1.0 s. Fivefold cross validation was used to test prediction. For each fold, the session's trials were randomly partitioned into five equally sized sets. Four of the five sets were used to train the GLM model and the remaining set was used to test. The prediction accuracy was calculated by dividing the number of correctly predicted trials by the total number of trials used for testing for each fold.

Significance was determined by performing the same procedure as described above but shuffling the trial outcomes to obtain a null distribution. At each time bin used for the prediction, a rank-sum test was performed on the prediction accuracy using the real data compared to the shuffled data.

## Occupancy maps

For all spatial field analyses, only the five animals that were trained on the full T-maze were used; the three animals trained on the truncated maze were excluded due to insufficient spatial data. Two-dimensional occupancy maps were generated by calculating occupancy in 2 cm square spatial bins from epochs in which the animals running speed exceeded 3 cm/s and convolving with a 2d Gaussian ($\sigma$=2 pixels).

## Place field determination

2-D occupancy-normalized firing rate maps were generated by dividing the spikes at each 2-D pixel by the unsmoothed occupancy at that pixel, and then smoothing with a Gaussian kernel of ($\sigma$=2 pixels, or 4 cm).

Linearized trajectory occupancy maps were calculated as previously described (*Jadhav et al., 2016*; *Karlsson and Frank, 2009*). Briefly, the rats 2d spatial coordinates were first segmented into run epochs based on contiguous bouts of time in which the animals running speed exceeded 3 cm/s. Then, the position of the animal during each 'run epoch' was categorized by its origin and destination, and each position was assigned to one of 100 (1.23 cm) bins from the origin to the destination of that route. Then the total number of spikes at each bin was divided by that bin's occupancy, and the map was convolved by a Gaussian kernel with a standard deviation of 2 pixels (4 cm).

Place field peak was calculated as the peak rate bin on the smoothed, linearized rate map, and place fields were only considered for trajectories in which the cell had a peak firing rate of at least 1 Hz. Place field width was calculated using a flood fill-algorithm in which the edges were defined as the closest bins to the peak bin in which the rate fell below 25% of the peak rate. Place field sparsity and information scores were calculated as previously described (*Skaggs et al., 1993*). A cell was determined to have a place field if its peak firing rate along that trajectory was ≥2 S.D. above the mean of a bootstrapped distribution generated by circularly shifting the spikes in time for each individual run, and if the field covered less than 75% of the linearized trajectory. When analyzing the place field characteristics of choice-selective cells (*Figure 7*, *Figure 7—figure supplement 1*), the field of each outbound journey was analyzed separately so as to prevent 'choosing' certain fields over others.

## Trajectory selectivity

Trajectory selective cells were identified using a previously validated method (*Shin et al., 2019*). Briefly, the linearized spatial tuning curve was calculated separately for each outbound trajectory and

the correlation between those trajectories was computed. A cell was identified as trajectory selective if this correlation value was lower than the 5th percentile of a distribution wherein each outbound trajectory identity was shuffled and the correlation recalculated. Thus, trajectory selectivity was deduced if there was an anticorrelation in the spatial patterns of firing between the two runs.

To measure trajectory selectivity as it related to decision period selectivity, we used an analogous method to the odor selectivity index. Briefly, we calculated the mean firing rate along each run beginning one half second following odor port exit and once the animal's speed exceeded 3 cm/s to the end of that run (when the animal either reached the goal or its velocity fell below 3 cm/s for more than ½ second). The difference in these mean firing rates across runs was divided by the sum of those two mean rates to generate a trajectory selectivity index.

## Bayesian decoder

Choice identity decoding was performed as previously described (*Karlsson and Frank, 2009*). Briefly, a memoryless Bayesian decoder was built for each of the two choice identities from spikes occurring in the odor sampling period. Then, the likelihood of each choice (x) was reconstructed from their posterior probabilities given the spikes occurring at each segment along the maze on each run (p(x | spikes)=p(spikes | x) * p(x) / p(spikes)). Run activity was defined as contiguous bouts following the odor sampling period when the animal was traveling above 3 cm/sec. Additionally, to prevent overlap between odor-period spiking and run spiking, the run periods were defined beginning 0.5 s following odor port disengagement time. We assumed that the N active cells fired independently and followed a Poisson process, giving the following equation.

$$P\left(X|spikes\right) = C * (\textstyle\prod_{i=1}^{N} f_i(X)^{spikes_i}) * e^{-\tau} \sum_{i=1}^{N} f(X)$$

where C is a normalization constant across the two choice identities. The p-value was calculated from the Gaussian statistics of a Monte Carlo random shuffle (200 shuffles) of choice identity during the odor sampling period. When choice identity decoding was calculated from the likelihoods of activity from each run, the decoding was performed in the same fivefold leave-one-out fashion as was used in the GLM analyses, and the p value was calculated from the same Monte Carlo random shuffle of the training set route identities as the odor period based decoding.

## Code Availability

All data processing, analyses and statistics in this study were conducted using open-source package Mountainsort (https://github.com/flatironinstitute/mountainsort), and custom code in MATLAB (R2018), unless otherwise noted. All custom code is available on GitHub at:, (copy archived at swh:1:rev:88d5f8d2bb39796b8656dc42bd42969a7bbe8697; *Bladon, 2022*).

## Acknowledgements

This work was supported by NIH Grants R01MH120228, R01MH112661, and a Smith Foundation Odyssey award to SPJ. We thank the late Howard Eichenbaum for assistance with the behavioral task.

## Additional information

### Funding

| Funder | Grant reference number | Author |
| --- | --- | --- |
| National Institute of Mental Health | R01MH120228 | Shantanu P Jadhav |
| National Institute of Mental Health | R01MH112661 | Shantanu P Jadhav |

The funders had no role in study design, data collection and interpretation, or the decision to submit the work for publication.

## Author contributions
Claire A Symanski, Data curation, Formal analysis, Validation, Investigation, Visualization, Methodology, Writing - original draft, Writing - review and editing; John H Bladon, Data curation, Software, Formal analysis, Validation, Visualization, Methodology, Writing - review and editing; Emi T Kullberg, Investigation, Methodology; Paul Miller, Formal analysis; Shantanu P Jadhav, Conceptualization, Software, Formal analysis, Supervision, Funding acquisition, Visualization, Methodology, Writing - original draft, Project administration, Writing - review and editing

## Author ORCIDs
John H Bladon http://orcid.org/0000-0001-8993-9898
Shantanu P Jadhav http://orcid.org/0000-0001-5821-0551

## Ethics
All experimental procedures were approved by the Brandeis University Institutional Animal Care and Usage Committee (IACUC) and conformed to US National Institutes of Health. Procedures were approved under IACUC Protocol # 21001.

## Decision letter and Author response
Decision letter https://doi.org/10.7554/eLife.79545.sa1
Author response https://doi.org/10.7554/eLife.79545.sa2

# Additional files

## Supplementary files
• Supplementary file 1. Average CA1 and PFC cell counts per session in each category. Overall averages are indicated as mean ± s.e.m.

• MDAR checklist

## Data availability
Data is available for download on figshare: Data DOI: https://doi.org/10.6084/m9.figshare.19620783.v1.

The following dataset was generated:

| Author(s) | Year | Dataset title | Dataset URL | Database and Identifier |
|---|---|---|---|---|
| Symanski CA, Bladon JH, Kullberg ET, Miller P, Jadhav SP | 2022 | Rhythmic coordination of hippocampal-prefrontal ensembles for odor-place associative memory and decision making | https://doi.org/10.6084/m9.figshare.19620783.v1 | figshare, 10.6084/m9.figshare.19620783.v1 |

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
