## [Editor Report]

The authors report coordination mechanisms between oscillations recorded in the CA1 subfield of the hippocampus, prefrontal cortex, and olfactory bulb and cell ensemble activity in CA1 and prefrontal cortex during odor-cued decision-making. The important findings support the hypothesis that the β rhythm plays a role in coordinating CA1-prefrontal cortex ensembles during decision-making. Sensory-guided decision-making is of broad significance to many readers who are studying executive functions and decision-making behaviors, and the observations reported in this manuscript provide convincing evidence of mechanisms that may support these functions and behaviors.

---

## [Decision Letter]

**Decision letter after peer review:**

Thank you for submitting your article "Rhythmic coordination of hippocampal-prefrontal ensembles for odor-place associative memory and decision making" for consideration by *eLife*. Your article has been reviewed by 3 peer reviewers, and the evaluation has been overseen by a Reviewing Editor and Laura Colgin as the Senior Editor. The following individual involved in the review of your submission has agreed to reveal their identity: Carmen Varela (Reviewer #1).

Essential revisions:

Essential revisions include additional data analyses and text revisions, as outlined in the individual reviews included in their entirety below.

*Reviewer #1 (Recommendations for the authors):*

I only have a few suggestions to improve clarity.

Provide a rationale for the truncated maze task in the text before figure 1 (it is mentioned in Figure 1b legend for the first time -and later in methods).

Line 139: state the metric of 'change' presented in Figure 2d,e; I think is the difference, or is it a percent change?

Line 189: clarify here or in methods if the un-cued, 'air' only, trials were interleaved with 'odor' trials and if those sessions were performed after the rat learned the task (if not the animal may not be sniffing or be motivated to try to discriminate).

Lines 251-253: Panel 3e,3i – describe what the bottom plots are (between 0-1s ---after the odor sampling window?).

Figure 6e: it would be helpful to include labels/titles on the left- and right-side panels to indicate task-responsive Pyr. versus all task-responsive cells (as indicated in the legend).

Figure 7a: change 'nosepoke' to 'odor port' for consistency with text.

Line 414: it may be helpful to indicate that odor sampling periods were excluded from spatial analysis earlier in the section.

Lines 445,448: Reference to the figure panels should be to Figure 7f in 445, and Figure 7g in line 448.

Lines 494-95: change to '…compared to pre-odor periods as well as compared to the period of immobility during reward consumption' to avoid confusion.

Line 876: Could you expand on how trajectory selective cells were identified? The methods state that a cell was identified as selective if the correlation between the spatial tuning curves for each outbound trajectory were higher than the null distribution, however, shouldn't the correlation between the two outbound trajectories be low (if a cell is selected for a specific trajectory compared to the other)? When looking at the methods in the Shin et al., 2019 referenced paper it wasn't clear which is/are the method/s that readers should look for; the 'trajectory dependent' metric in Shin et al. seems to correspond to the calculation of the trajectory selectivity index, but not sure how the cells were identified as selective (first paragraph in Methods-trajectory selectivity).

*Reviewer #2 (Recommendations for the authors):*

1) To address the first concern in the public review, I recommend that the authors check the phase difference of β oscillations in all three region pairs. The authors might also consider investigating how neurons show different degrees of phase-locking to β in the OB, mPFC, and CA1.

2) In addition, the abstract shows "with CA1-PFC β and RR coherence enhanced during the odor-cued decision-making period. (lines 7-8)." Although this is not inaccurate, it may give readers the wrong impression that only CA1-PFC coupling was enhanced during decision-making. I suggest rewriting this sentence to clarify that the coherence of all pairs was elevated. Alternatively, the authors may wish to include additional data suggesting that CA1-PFC coupling shows better correlations with task phase/performance than CA1-OB and PFC-OB coupling, but this is unnecessary if the relevant text is reworded appropriately.

3) For the second point, Figure 5g can be expanded to include the preferred phase of mPFC pyramidal and interneurons. Also, Figure 5g does not seem to be explained in the Results section.

4) Please clarify the difference between the tile plots shown on the top and bottom in Figure 3e and i.

5) Some paragraphs in the result section (Line 283-291 and Line 345-354) seem more like a discussion than descriptions of results. The authors might consider moving these paragraphs to the Discussion section.

*Reviewer #3 (Recommendations for the authors):*

I pasted what I consider as main points in the public review.

Regarding point (A), perhaps the authors could bring some of the supplementary figures related to oscillations as the main ones, as well as perform an analysis trying to link population spiking activity to the observed rhythms (c.f. point B). Further oscillatory analyses (e.g., directionality between regions, search for oscillatory activity in the spike times) could also be performed to let the presentation of the results more "oscillation-centered" in a way of matching the title and main text.

Regarding point (C), I feel that it is my duty to inform the editor that I detected an overlap between these findings and previous research, and she can make the novelty judgment (I actually favor important results being corroborated by independent labs). In any case, perhaps the authors want to make a stronger case in their text to defend this point. Regarding (D), the authors could clarify any misinterpretation of mine about the small percentage of detected cells potentially not being above chance level, or else perhaps use a statistically corrected threshold against false positives to classify the neurons.

---

## [Author Response]

Essential revisions:Essential revisions include additional data analyses and text revisions, as outlined in the individual reviews included in their entirety below.

We thank the Reviewers for their constructive feedback and comments, which have enabled us to substantially improve the manuscript. The Reviewers appreciated the significance of the results, and made several suggestions for additional analyses and interpretation of the results, which are addressed below in the individual responses. Reviewer 3 also raised points about framing our results in the context of known increases in β coherence in odor-sampling tasks. We have added additional analyses that strengthen our results with regards to rhythmic coordination and activity dynamics during odor-cued spatial decision making, and emphasized the novelty of our results and the advance that they provide.

Response to the Public Reviews and a detailed point-by-point response that addresses all of the Reviewer’s comments is included below.

Reviewer #1 (Recommendations for the authors):I only have a few suggestions to improve clarity.Provide a rationale for the truncated maze task in the text before figure 1 (it is mentioned in Figure 1b legend for the first time -and later in methods).

We thank the Reviewer for this comment. We now clarify and provide a rationale for the truncated maze in the text before Figure 1 in addition to the Methods section. The truncated maze was a stage in the training procedure, and we included 3 animals who only ran this truncated maze in relevant analyses, specifically related to the decisionmaking period during odor-sampling period.

– Page 6 line 92 to line 97 include text regarding the truncated maze used for animals 6-8

– Methods were clarified as well, page 33, lines 684-687

Line 139: state the metric of 'change' presented in Figure 2d,e; I think is the difference, or is it a percent change?

The metric is difference, we have now added this metric in the caption for Figure 2d,e, as well as in the y-axis of the figure and in the methods (Page 36, lines 753-755).

Line 189: clarify here or in methods if the un-cued, 'air' only, trials were interleaved with 'odor' trials and if those sessions were performed after the rat learned the task (if not the animal may not be sniffing or be motivated to try to discriminate).

We have revised the uncued air sessions methods section to clarify that all trials were uncued in the ‘air’ only sessions, and that the sessions were performed after the rat learned the task. (Page 33 lines 699-702)

We have also added context in the Results section for the uncued air sessions (Page 10, lines 196-198)

Lines 251-253: Panel 3e,3i – describe what the bottom plots are (between 0-1s ---after the odor sampling window?).

We have reorganized the plots to be more legible, the plots now have revised x-axes at left (locked to odor onset) and right (locked to offset). We also added text on page 1213, lines 258-261 to indicate left and right panels now in Figure 3e,f; namely that they are now in alignment to odor onset and offset.

Figure 6e: it would be helpful to include labels/titles on the left- and right-side panels to indicate task-responsive Pyr. versus all task-responsive cells (as indicated in the legend).

We have added labels to indicate task-response pyramidal cells.

Figure 7a: change 'nosepoke' to 'odor port' for consistency with text.

We have changed all references of ‘nosepoke’ to odor port for consistency with text.

Line 414: it may be helpful to indicate that odor sampling periods were excluded from spatial analysis earlier in the section.

We have added this clarification earlier in the section, on page 20, line 424.

Lines 445,448: Reference to the figure panels should be to Figure 7f in 445, and Figure 7g in line 448.

Thank you pointing this out. We have updated the figure panels (now 7e and 7f) and have corrected the references (now on page 21 line 464 and line 467).

Lines 494-95: change to '…compared to pre-odor periods as well as compared to the period of immobility during reward consumption' to avoid confusion.

We have made this change (now on page 24 line 520-521).

Line 876: Could you expand on how trajectory selective cells were identified? The methods state that a cell was identified as selective if the correlation between the spatial tuning curves for each outbound trajectory were higher than the null distribution, however, shouldn't the correlation between the two outbound trajectories be low (if a cell is selected for a specific trajectory compared to the other)? When looking at the methods in the Shin et al., 2019 referenced paper it wasn't clear which is/are the method/s that readers should look for; the 'trajectory dependent' metric in Shin et al. seems to correspond to the calculation of the trajectory selectivity index, but not sure how the cells were identified as selective (first paragraph in Methods-trajectory selectivity).

Thank you for pointing this out. This was an error in text, and the text now reads that the correlation between the two outbound trajectories were *lower* than the null distribution. We have revised the subsection trajectory selectivity to clarify that we were searching for an anticorrelation between the spatial patterns of firing.

We have added on page 43 lines 920-922:

“Thus, trajectory selectivity was deduced if there was an anticorrelation in the spatial patterns of firing between the two runs.”

Reviewer #2 (Recommendations for the authors):1) To address the first concern in the public review, I recommend that the authors check the phase difference of β oscillations in all three region pairs. The authors might also consider investigating how neurons show different degrees of phase-locking to β in the OB, mPFC, and CA1.

We thank the Reviewer for this suggestion. As the Reviewer noted, we do not interpret this result as exclusively PFC-CA1 interactions; instead as an OB-CA1-PFC interacting network coordinated by oscillations during the odor task, which is now emphasized in the abstract and text.

We show β synchrony across all three regions, OB, CA1 and PFC in Figure 2, and have added analyses that shows different degrees of phase-locking in Figure 5 and Figure 5—figure supplement 1. The phase-locking results indicate that the degree of phase-locking for CA1 and PFC neurons is stronger for CA1 or PFC β than to OB β (Figure 5d, Figure 5—figure supplement 1b, i, j), suggesting that this phase coherent firing is likely not due to common inputs from OB. In particular, CA1 interneurons phase-lock strongly to both CA1 β and cross-regional PFC β during correct vs. incorrect trials and this was not true for OB β, suggesting a key role of CA1 interneurons in the PFC-CA1 β interactions.

We have added on page 17 lines 369-372:

“Interestingly, cross-region spike-phase coherence showed a similar relationship to decision accuracy for CA1 phase-coherent interneurons… … suggesting a central role for CA1 interneurons in synchronizing PFC-CA1 activity and enabling a correct choice.”

The phase difference of β and RR oscillations across regions showed high variability across animals and sessions (Figure 2—figure supplement 2d), presumably due to differences in electrode locations across animals, especially in PFC and OB (Figure 2figure supplement 1a). We see a slight preference for CA1 leading PFC for β and RR rhythms, and it is likely that CA1 interneurons may play a role in this rhythmic interaction based on phase-locking and cross-correlation results. However we refrain from a strong claim of directionality of interaction using phase difference due to this variability.

We have added on page 11 lines 212-217:

“We also examined phase differences in β and RR oscillations between all pairs of regions during decision-making…”

And in the Discussion section, page 23-24 lines 517-519:

“There remained a consistent β-phase and RR phase offset between PFC and CA1 during odor sampling, suggesting a subtler interaction between regions at those frequencies.”

2) In addition, the abstract shows "with CA1-PFC β and RR coherence enhanced during the odor-cued decision-making period. (lines 7-8)." Although this is not inaccurate, it may give readers the wrong impression that only CA1-PFC coupling was enhanced during decision-making. I suggest rewriting this sentence to clarify that the coherence of all pairs was elevated. Alternatively, the authors may wish to include additional data suggesting that CA1-PFC coupling shows better correlations with task phase/performance than CA1-OB and PFC-OB coupling, but this is unnecessary if the relevant text is reworded appropriately.

We thank the Reviewer for this suggestion. We have now updated the abstract to indicate that coherence across all regions was enhanced during decision-making, as shown in Figure 2.

We note in the Abstract:

“During odor sampling, the β (20-30 Hz) and respiratory (7-8 Hz) rhythms (RR) were prominent across the three regions, with β and RR coherence between all pairs of regions enhanced during the odor-cued decision making period.”

3) For the second point, Figure 5g can be expanded to include the preferred phase of mPFC pyramidal and interneurons. Also, Figure 5g does not seem to be explained in the Results section.

We thank the Reviewer for this suggestion. We have included the preferred phase of mPFC pyramidal and interneurons for β and RR rhythms in Figure 5. The β phase of PFC pyramidal neurons and interneurons did not show clear phase preferences that indicate temporal relationship within β oscillations, which is noted in the text. Nonetheless, the cross-correlation results in Figure 4 that suggest temporal relationships during decision making, CA1 interneuron phase preferences for both CA1 and PFC β, and the change in β phase-locking for correct and incorrect trials, together support the view that β rhythms play a role in coordinating activity.

We therefore emphasize changes in phase preference for correct vs. incorrect trials in Figure 5. We have updated Figure 5 and Figure 5—figure supplement 1, and now have references to all panels in Figure 5 in the text.

4) Please clarify the difference between the tile plots shown on the top and bottom in Figure 3e and i.

We thank the Reviewer for pointing this out, similar to Reviewer 1. The top and bottom plots were aligned either to odor onset or to odor offset. We have now revised these plots to be side-by-side and read more intuitively (now panels Figure 3e and 3f).

5) Some paragraphs in the result section (Line 283-291 and Line 345-354) seem more like a discussion than descriptions of results. The authors might consider moving these paragraphs to the Discussion section.

We have updated the manuscript in a number of places to move these paragraphs to the Discussion section.

Reviewer #3 (Recommendations for the authors):I pasted what I consider as main points in the public review.Regarding point (A), perhaps the authors could bring some of the supplementary figures related to oscillations as the main ones, as well as perform an analysis trying to link population spiking activity to the observed rhythms (c.f. point B). Further oscillatory analyses (e.g., directionality between regions, search for oscillatory activity in the spike times) could also be performed to let the presentation of the results more "oscillation-centered" in a way of matching the title and main text.Regarding point (C), I feel that it is my duty to inform the editor that I detected an overlap between these findings and previous research, and she can make the novelty judgment (I actually favor important results being corroborated by independent labs). In any case, perhaps the authors want to make a stronger case in their text to defend this point. Regarding (D), the authors could clarify any misinterpretation of mine about the small percentage of detected cells potentially not being above chance level, or else perhaps use a statistically corrected threshold against false positives to classify the neurons.

We have addressed these main points in the public review.